# HIV-1 Vpr combats the PU.1-driven antiviral response in primary human macrophages

Maria C. Virgilio [1,2,3], Barkha Ramnani [3], Thomas Chen[2,3,4], W. Miguel Disbennett[4,5,7], Jay Lubow[4,8], Joshua D. Welch [2,6] & Kathleen L. Collins [1,3,4] ✉

HIV-1 Vpr promotes efficient spread of HIV-1 from macrophages to T cells by transcriptionally downmodulating restriction factors that target HIV-1 Envelope protein (Env). Here we find that Vpr induces broad transcriptomic changes by targeting PU.1, a transcription factor necessary for expression of host innate immune response genes, including those that target Env. Consistent with this, we find silencing PU.1 in infected macrophages lacking Vpr rescues Env. Vpr downmodulates PU.1 through a proteasomal degradation pathway that depends on physical interactions with PU.1 and DCAF1, a component of the Cul4A E3 ubiquitin ligase. The capacity for Vpr to target PU.1 is highly conserved across primate lentiviruses. In addition to impacting infected cells, we find that Vpr suppresses expression of innate immune response genes in uninfected bystander cells, and that virion-associated Vpr can degrade PU.1. Together, we demonstrate Vpr counteracts PU.1 in macrophages to blunt antiviral immune responses and promote viral spread.

The HIV-1 genome encodes several accessory proteins that counteract innate and adaptive antiviral responses. Although HIV accessory proteins have been widely studied, the role of the Vpr accessory protein remains enigmatic. Vpr is highly conserved amongst lentiviruses and is necessary for optimal replication in macrophages[1]. In addition, it is unique amongst HIV accessory proteins in that it is packaged in the virus particle at high levels through specific interactions with Gag p6[2]. Studies using human lymphoid tissue, which is rich in both T cells and macrophages, have shown that loss of Vpr decreases virus production, but only when the virus strain is capable of efficiently infecting macrophages[1,3–6]. These studies provide evidence that Vpr enhances infection of macrophages and increases viral burden in tissues containing macrophages. Vpr localizes to the nucleus[7], where it induces G2 cell cycle arrest by targeting a variety of host factors involved in post-replication DNA repair[8–15]. To target these factors, Vpr requires the host protein DCAF1 (also known as Vpr binding protein (VprBP))[16]. Through DCAF1, Vpr interacts with damaged DNA binding protein 1

(DDB1) as part of the Cul4A E3 ubiquitin ligase complex where host proteins recruited by Vpr are ubiquitylated and degraded[17–22]. However, the effects of Vpr on DNA repair and cell cycle arrest do not provide a clear explanation for how Vpr enhances HIV infection of macrophages.

Macrophages are terminally differentiated antigen-presenting cells that are critical for many immune functions, including antiviral innate immune responses[23,24]. Macrophage identity is tightly controlled through the timed expression of myeloid transcription factors, particularly PU.1 (reviewed in ref. 25). PU.1 is a hematopoietic-specific and ETS family transcription factor that is essential for lymphoid and myeloid development[25–28]. ETS family proteins bind to purine-rich DNA domains with a central GGAA/T core consensus[29]. Macrophages require early and continuously high levels of PU.1 expression (reviewed in refs. 25,26) to maintain normal functionality. PU.1 regulates many essential macrophage genes, including those for cytokine receptors M-CSF and GM-CSF and the adhesion molecule CD11b. PU.1 also

[1]Cellular and Molecular Biology Program, University of Michigan, Ann Arbor, MI, USA. [2]Department of Computational Medicine and Bioinformatics, University of Michigan, Ann Arbor, MI, USA. [3]Department of Internal Medicine, University of Michigan, Ann Arbor, MI, USA. [4]Department of Microbiology and Immunology, University of Michigan, Ann Arbor, MI, USA. [5]Post-Baccalaureate Research Education Program (PREP), University of Michigan, Ann Arbor, MI, USA. [6]Department of Computer Science and Engineering, University of Michigan, Ann Arbor, USA. [7]Present address: University of Pennsylvania, Philadelphia, PA, USA. [8]Present address: ImmunoVec, Inc., Los Angeles, CA, USA. ✉e-mail: klcollin@umich.edu

coordinates with other transcription factors such as IRF4 and C/EBPα, and TET methyl cytosine dioxygenase 2 (TET2; also known as ten-eleven translocation 2) to regulate gene expression[25,30–32].

We previously reported that Vpr counteracts accelerated degradation of the HIV Env protein, which occurs in HIV-infected macrophages but not T cells[33,34]. We hypothesized that Vpr disables a macrophage-specific restriction factor that detects and degrades HIV Env. We recently identified this factor as the macrophage mannose receptor (MR)[35], which is highly expressed in macrophages but not T cells[36,37]. MR senses HIV Env via densely packed high mannose residues that serve as pathogen-associated molecular patterns (PAMPs) because they are normally absent from host cellular proteins. MR recognition of HIV Env disrupts infection by promoting lysosomal degradation of Env and Env-containing viral particles[33,35]. Although the canonical targeting of host factors by Vpr involves the binding of Vpr to DCAF1, leading to ubiquitylation and degradation via the proteosome, Vpr does not directly interact with MR[35]. Instead, Vpr suppresses expression of the MR gene, *MRC1*[35].

In this study, we examine the mechanism by which Vpr suppresses the expression of *MRC1* and demonstrate that Vpr additionally exerts suppressive effects on other important genes in macrophages, enhancing virion assembly and spread. Single-cell RNA sequencing (scRNA-seq) of primary human macrophages infected with infectious wild type (WT) or Vpr-*null* HIV allowed us to distinguish the effects of Vpr on both infected and virally exposed, uninfected cells in the same culture. Within infected cells, Vpr selectively downregulated genes controlled by the macrophage-selective transcription factor, PU.1. By targeting PU.1, Vpr systemically disrupted several antiviral factors, including *MRC1*, and *IFITM3*, a previously reported target of Vpr that is also capable of disrupting Env function[38]. Consistent with this, silencing PU.1 in macrophages infected with Vpr-*null* virus rescued Env production. Vpr caused a systemic reduction of genes implicated in Toll-like receptor (TLR) and type I interferon (IFN-I) signaling that affected all the cells in the culture. Thus, we provide a systems level explanation for the positive effect of Vpr on HIV spread in macrophages. Finally, we found that Vpr-mediated downmodulation of PU.1-regulated gene expression is mediated by protein-protein interaction between Vpr and PU.1, resulting in accelerated proteasomal degradation of PU.1. This activity of Vpr is conserved amongst all HIV-1 molecular clones tested as well as HIV-2 and SIV. Remarkably, both the interaction between PU.1 and Vpr as well PU.1's subsequent degradation requires the Vpr interacting protein, DCAF1. The PU.1 transcriptional co-factor, TET2, is co-recruited with PU.1 to DCAF1 by Vpr. This aligns our results with other reports that Vpr targets TET2[38,39]. Together, our data support a model in which Vpr promotes HIV spread via systemic detrimental effects on the host innate antiviral response to infection.

## Results

### Single-cell RNA sequencing of HIV-1 infected MDMs reveals Vpr-dependent transcriptional changes

Vpr counteracts a number of host factors, primarily those implicated in DNA repair and restriction of HIV Env[35,38]. Recently we demonstrated that Vpr decreases transcriptional expression of the host restriction factor, mannose receptor (*MRC1*), which we hypothesize results from Vpr targeting a macrophage-specific transcription factor (Fig. 1A). To identify the transcription factor targeted by Vpr, we undertook an unbiased approach to characterize the genome-wide effects HIV-1 Vpr in monocyte derived macrophages (MDMs). For these studies, we used an infectious HIV molecular clone (89.6) that has or lacks an intact *vpr* gene (Fig. 1B). We infected primary human CD14⁺ MDMs from three independent healthy human donors and allowed the infection to spread for 10 days before harvesting cells (uninfected, 89.6$^{wt}$ or 89.6$^{Δvpr}$ infected) and prepared them for single-cell gene expression analysis (Fig. 1C). LIGER (linked inference of genomic experimental relationships)[40] was used to integrate all nine data sets. This analysis facilitated alignment despite significant transcriptomic differences between donors and sample types. Uniform manifold approximation and projection (UMAP) visualization of all donor sets showed no significant donor or batch-specific differences amongst the samples (Fig. 1D). Based on gene expression patterns in each of the clusters, we determined the main cluster to be pro-inflammatory macrophages (Cluster 0), and we identified a minor population of cycling cells (Cluster 1) within the virus-treated cells (Fig. 1E, F). These two clusters do not appear to have an infection phenotype because cells from cultures exposed to HIV were distributed across both the clusters (Fig. 1E, G).

To separate the bona fide infected cells from bystander cells, we computationally segregated cells into a subset that expressed both *tat* and *gag* transcripts from those that expressed neither (Fig. 1H, Supplementary Table 1). From a total of 13,639 WT-Vpr and 35,780 Vpr-*null* exposed MDMs, 6156 and 8699 were identified as *gag*⁺/*tat*⁺, respectively. The proportion of infected cells identified by donor using this analysis was similar to that found by the standard method of identifying infected cells by intracellular Gag staining and detection by flow cytometry (Supplementary Table 2). A comparison of gene expression profiles in cells with and without Vpr revealed that Vpr expression boosted HIV gene expression as previously reported[41] and caused a significant transcriptional shift in host gene expression (Fig. 1I). Using a two-sided Wilcoxon rank-sum test and false discovery rate correction, we identified 3150 genes with statistically significant two-fold or greater differential expression between 89.6$^{wt}$ and 89.6$^{Δvpr}$-infected cells.

### Vpr-repressed genes in MDMs include targets of the transcription factor PU.1

To identify the transcription factor(s) targeted by Vpr, we used the DNA motif identification software, HOMER (hypergeometric optimization of motif enrichment), to scan for common transcription factor-binding motifs in the promoters of genes downregulated in the presence of Vpr. Our analysis revealed several candidate transcription factors (TFs) and TF-binding families, including many members of the ETS family of TFs (Fig. 2A; Supplementary Fig. 1A). However, only PU.1 (encoded by the gene *SPI1*) was highly expressed in our data sets (Fig. 2B). Moreover, PU.1 appeared twice in the list – as both a member of the ETS family and as a co-factor in the ETS/IRF (interferon regulatory factor) family of transcription factors, a relationship that is well documented[30] (Fig. 2A). PU.1 is the master myeloid transcription factor (reviewed in refs. 27,42,43). It is required for terminal differentiation of macrophages and is necessary to maintain macrophage immune function[25]. From the list of most downregulated genes in Vpr-expressing MDMs, HOMER identified 316 genes with a PU.1 binding motif in their promoter and 670 PU.1-IRF co-regulated genes, for a combined 840 distinct PU.1-regulated genes identified (Fig. 2C, E). One of the genes identified with a PU.1 binding motif was interferon-induced transmembrane protein 3 (*IFITM3*). IFITM3 is an important host restriction factor that is similar to mannose receptor (MR), in that IFITM3 targets HIV Env and has reduced gene expression (by reverse transcription quantitative PCR (RT-qPCR)) in the presence of Vpr in HIV-infected MDMs[38]. Gene ontology analysis of the 840 genes targeted by PU.1 revealed that they are enriched for genes involved in regulation of the immune system and defense response (Fig. 2D).

Because we had previously demonstrated that the gene encoding mannose receptor (*MRC1*) was transcriptionally downmodulated approximately two-fold by Vpr in HIV-infected primary human macrophages using RT-qPCR[35], we examined our data set to specifically identify *MRC1* transcripts. In the scRNA-seq data set shown here, we identified a Vpr-dependent change in *MRC1* transcripts (log2FC = 0.932) that is statistically significant (false-discovery rate corrected two-sided Wilcoxon rank-sum p-value = 0.002) but fell just below our

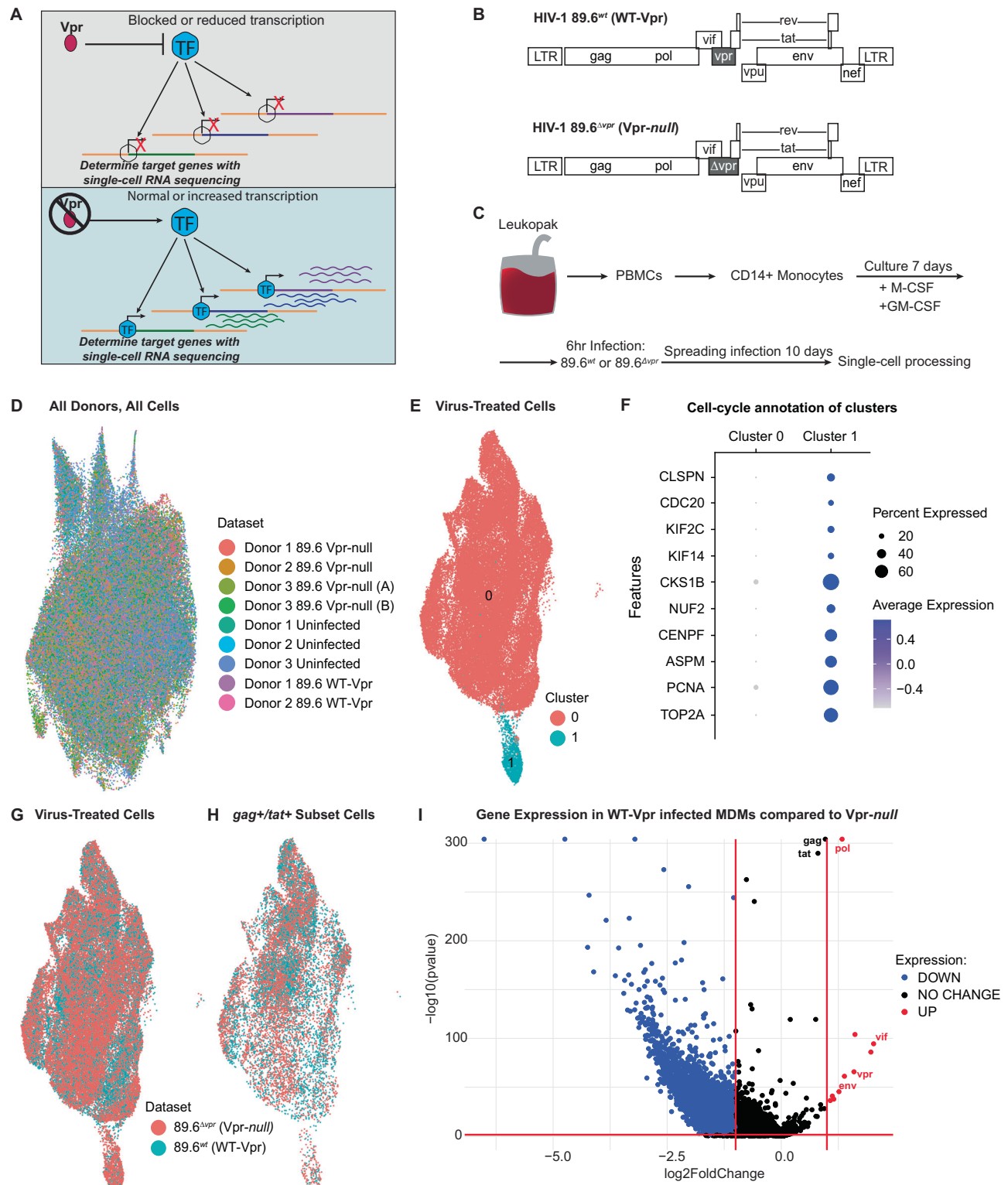

arbitrary cutoff of a two-fold change (Figs. 1H and 2C). Consistent with the pattern we observed for other Vpr-suppressed genes, we identified several PU.1 binding sites in the promoter region of *MRC1* (Supplementary Fig. 2A) when we manually scanned for PU.1 binding motifs, some of which have been previously described[44]. Furthermore, MDMs transduced with a short hairpin RNA (shRNA)-expressing lentivirus targeting *SPI1* transcripts reduced both PU.1 and MR protein (Supplementary Fig. 2B). While the effect of Vpr on *MRC1* transcriptional expression is modest, it combines synergistically with HIV Nef-

dependent disruption of MR trafficking, reversing mannose receptor-dependent lysosomal degradation of Env in HIV-infected primary macrophages[35].

## Vpr suppresses PU.1 regulated genes implicated in Toll-like receptor and IFN-I responses

PU.1 regulated genes that are downmodulated in Vpr-expressing cells include several factors implicated in Toll-like receptor (TLR) signaling pathways (Fig. 3A). TLRs are evolutionarily conserved, pattern

**Fig. 1 | Single-cell RNA sequencing of HIV-1 infected MDMs reveals Vpr-dependent transcriptional changes. A** Schematic diagram illustrating the objective of single-cell RNA sequencing; the identification of Vpr-targeted transcription factor(s) in HIV-infected primary macrophages; TF transcription factor. **B** Genome maps for full-length 89.6$^{wt}$ HIV-1 (top) and the same viral genome with a premature stop-codon in *upr* (bottom). **C** Experimental setup for the generation of the scRNA-seq datasets. **D** UMAP representation of LIGER-integrated scRNA-seq data from MDM samples treated as shown in **C** and listed in Dataset. **E, G, H** UMAP representations of LIGER-integrated scRNA-seq data from MDMs treated as indicated. **E** Colors indicate individual clusters. Cluster 0 = pro-inflammatory macrophages, Cluster 1 = cycling cells. **F** Dot plot representation of cell-cycle genes used to determine clusters in **E**. The size of the dot equates to the percentage of cells within the population expressing the feature, and the color indicates the average expression of the feature across all cells in each cluster. **G, H** Cells are colored according to whether they were exposed to 89.6 WT-Vpr virus (blue) or Vpr-*null* virus (pink). **H** Bona fide infected cells were identified based on expression of HIV *tat* and *gag*. **I** Volcano plot of differentially expressed genes from HIV 89.6$^{wt}$ verses 89.6$^{Δvpr}$ infected MDMs from **H** as determined by two-sided Wilcoxin Rank Sum. Significance determined as >1 log2 fold change and false discovery rate adjusted *p*-value of $p < 0.05$ (red-bars). Blue colored genes indicate genes less highly expressed in HIV 89.6$^{wt}$ verses 89.6$^{Δvpr}$ infected MDMs, red colored genes indicate genes more highly expressed, and black colored genes indicate no significant difference between datasets.

recognition receptors (PRRs) that recognize PAMPs ([45–47]), including HIV. To determine whether these effects of Vpr are restricted to infected cells within the culture, we compared gene expression profiles amongst all the sample types collected from our single-cell analysis: uninfected (virus naïve), 89.6$^{wt}$ and 89.6$^{Δvpr}$-infected, and 89.6$^{wt}$ and 89.6$^{Δvpr}$ virally exposed but uninfected (bystander) cells. This was accomplished by computationally segregating cells into subsets that expressed both *tat* and *gag* transcripts from those that were virally exposed but appeared uninfected (*tat* and *gag*-negative) (Fig. 3B).

As expected, expression of genes involved in the TLR-mediated IFN-I response to infection, including interferon stimulated gene 15 (*ISG15*), which is upregulated in response to IFN-I and TLR-signaling (reviewed in refs. [48,49]), *LY96*, and interferon-inducible protein-6 (*IFI6*) (PU.1-IRF regulated genes) and *IFITM3* were very low in uninfected, unexposed cells (Fig. 3C). In contrast, each of these genes was upregulated in cells from HIV-treated primary macrophage cultures (Fig. 3C). As discussed above, Vpr counteracted upregulation of this antiviral response in infected cells (Figs. 2C, E and 3C). In addition, we were surprised to find many of these genes were upregulated in bystander cells as well (Fig. 3C) and that Vpr limited their induction. For example, we found that *ISG15* was more highly expressed in 89.6$^{Δvpr}$ bystander cells compared to 89.6$^{wt}$-exposed bystander cells. Similar observations were made for *LY96*, *IFI6*, and *IFITM3*. Interestingly, the protein product for *LY96* is myeloid differentiation 2 (MD2), which is required for TLR4 ligand-induced activation at the cell surface[50,51]. By comparison, other subsets of genes including *SPI1* and *MRC1* were primarily downmodulated in the infected subset and either not at all in the bystander cells or to a lesser extent than their infected counterparts (Fig. 3B).

Next, we investigated whether we could observe a Vpr-dependent suppression of PU.1-regulated gene products involved in the TLR and IFN-I signaling pathways using confocal fluorescent microscopy, which allowed us to distinguish between infected and uninfected cells within the same culture of MDMs, analogous to our scRNA-seq experiments. Quantification of ISG15 and IFITM3 in Gag+ cells showed significant reduction of both PU.1 gene products in HIV-infected MDMs expressing Vpr relative to cells infected with a Vpr-*null* HIV (Fig. 3E, H). Strikingly, we also observed a similar pattern in virus-exposed but uninfected MDMs (Fig. 3F, I), where uninfected cells exposed to wild type HIV had lower levels of PU.1 gene products compared to uninfected cells exposed to Vpr-*null* HIV. These results are consistent with the scRNA-seq data from Fig. 3C, confirming Vpr-dependent downmodulation of PU.1 regulated genes and their protein products in wild type infected MDMs as well as bystander cells in the same culture.

**PU.1 protein levels decrease in the presence of Vpr**
Vpr acts as an adapter protein that links host proteins to the Cul4A E3 ubiquitin ligase complex for ubiquitylation and degradation[17,18]. Thus, we hypothesized that Vpr downregulates PU.1-regulated genes in infected cells by targeting PU.1 for proteasomal degradation. To test our hypothesis, we first investigated whether Vpr reduced PU.1 protein levels in infected MDMs. We treated macrophages with a VSV-G

envelope-pseudotyped replication-defective clone of HIV NL4-3 that expresses GFP in the *env* reading frame (Fig. 4A). We transduced MDMs with this virus containing all intact accessory proteins, or with additional mutations in the open reading frames of either *upr*, *nef*, or both. We measured PU.1 levels by flow cytometry at 5-, 7-, and 10-days post infection. Notably, we found that Vpr expression resulted in lower PU.1 protein levels in infected (GFP+) cells whether Nef was expressed or not (Fig. 4B, C). These changes in PU.1 were consistently observed in MDMs from four independent donors (Fig. 4C). Based on these results, we concluded that Vpr downmodulates PU.1 in HIV-infected primary human MDMs. Under the conditions of this assay, in which MDMs were treated with a replication-defective virus and analyzed at least five days post infection, we did not observe a significant Vpr-dependent downmodulation of PU.1 in uninfected (GFP-) bystander cells (Supplementary Fig. 3A).

To determine whether Vpr was sufficient for PU.1 downmodulation, we tested the ability of several 3xFLAG-tagged Vpr proteins derived from a panel of HIV molecular clones [89.6, NL4-3, AD8 and YU2-Vpr (Fig. 4D)] to reduce exogenous PU.1 expression in transfected HEK 293T cells. Because the gene that encodes PU.1 (*SPI1*) contains PU.1 binding sites and is downmodulated by Vpr expression (Fig. 2B), for these experiments PU.1 was expressed under the control of a heterologous promoter that allowed separation of transcriptional and post-transcriptional changes (Fig. 4E). For comparison, we used the evolutionarily related protein, Vpx, from the HIV2$_{ROD}$ molecular clone. We found that all Vpr-expression constructs tested resulted in a notably lower level of PU.1 protein compared to the level of PU.1 protein with HIV-2$_{ROD}$Vpx (Fig. 4F). Vpr-dependent reduced levels of PU.1 were consistently observed in four independent experiments and were statistically significant (Fig. 4G).

After determining that Vpr was sufficient to downmodulate PU.1 in HEK 293T cells, we confirmed these results in a second cell line, the lymphoblastic chronic myelogenous leukemia cell line K562, which endogenously expresses PU.1. For these experiments Vpr-expression was achieved using VSV-G envelope pseudotyped lentiviruses. Like the results obtained in HEK 293T cells, infections with viruses expressing all Vprs tested (89.6, NL4-3, AD8, YU2) had lower endogenous PU.1 levels compared to mock infection and HIV-2$_{ROD}$Vpx infection (Fig. 4H).

Because the Vpr-dependent decrease of PU.1 occurred both for endogenous PU.1 expressed from its native promoter in K562 cells and in HEK 293T cells when PU.1 was expressed from a heterologous promoter, we hypothesized that Vpr was affecting PU.1 protein levels post-transcriptionally. To rule out transcriptional effects on the heterologous promoter, we quantified PU.1 mRNA in HEK 293T cells transfected with a PU.1 expression plasmid and HIV constructs derived from 89.6 that had or lacked the Vpr gene (Supplementary Fig. 4A). As shown in Fig. 4I, Vpr did not reduce expression of PU.1 mRNA when expressed from a heterologous promoter in transfected cells. Thus, we concluded that the mechanisms by which Vpr reduced PU.1 protein in HEK 293T cells must be post-transcriptional. Moreover, in macrophages Vpr likely exerts negative effects directly on PU.1 protein and

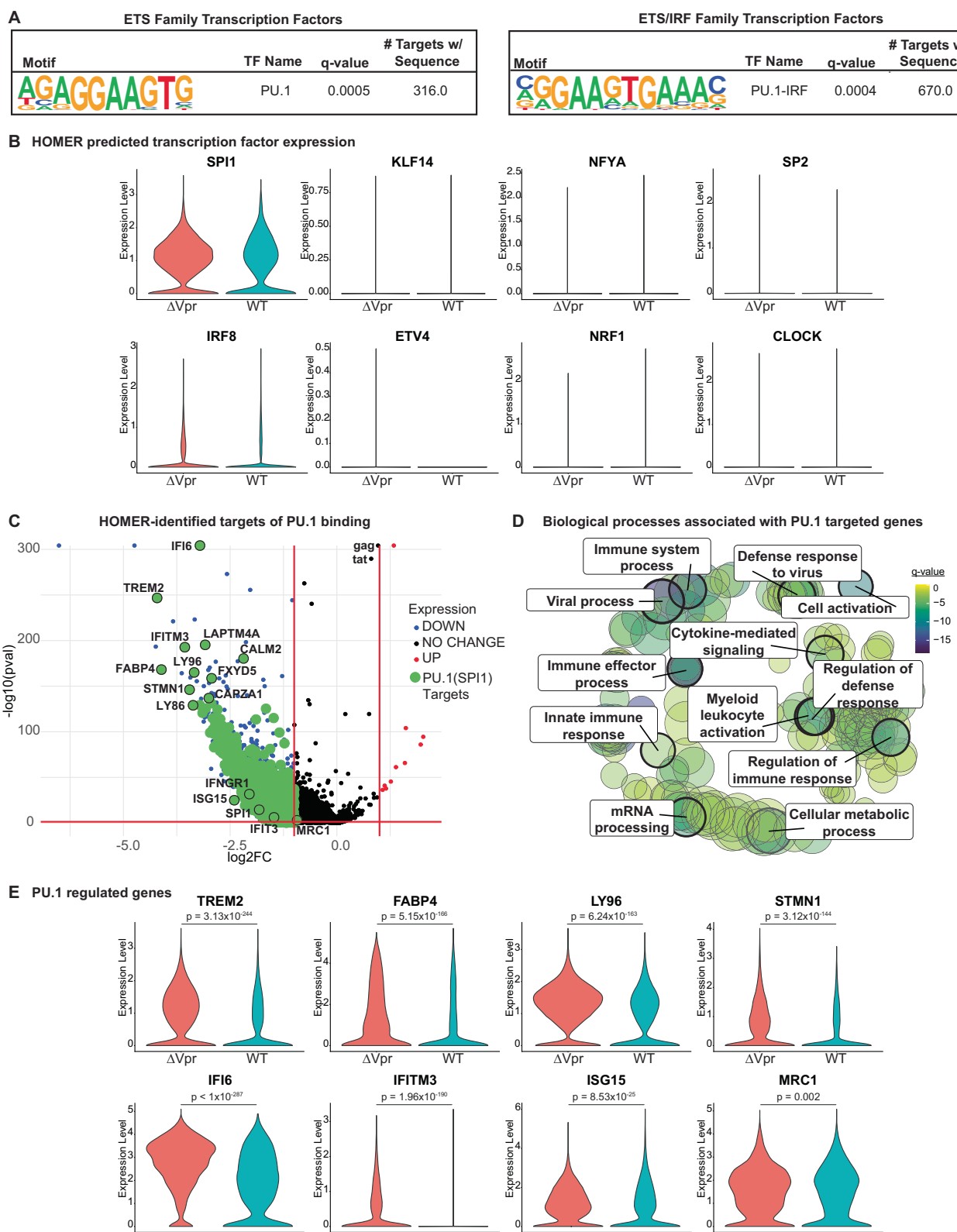

**Fig. 2 | Vpr downmodulates PU.1-dependent transcription. A** PU.1 motifs identified by HOMER as present in the promoters of Vpr-downmodulated genes (Fig. 1I, Blue). **B** Violin plots displaying RNA abundance of the indicated transcription factor genes in MDMs infected with the indicated virus. **C** Volcano plot as in Fig. 1I except that genes containing a PU.1 or PU.1-IRF binding motif in their promoter region are highlighted in green (see also Supplementary Fig. 2). **D** Biological processes associated with the PU.1 targeted genes from **C**. Size of circles indicates the relative

number of GO terms associated with the process. FDR adjusted *q*-values associated with GO terms are indicated by the color. Bolded rings are associated with biological processes listed. **E** Violin plots displaying RNA abundance of the indicated genes in MDMs infected with the indicated virus. HIV-1 89.6$^{wt}$ (WT); HIV-1 89.6$^{\Delta vpr}$ (ΔVpr). False-discovery rate corrected two-sided Wilcoxon rank-sum p-values are shown above the conditions being compared.

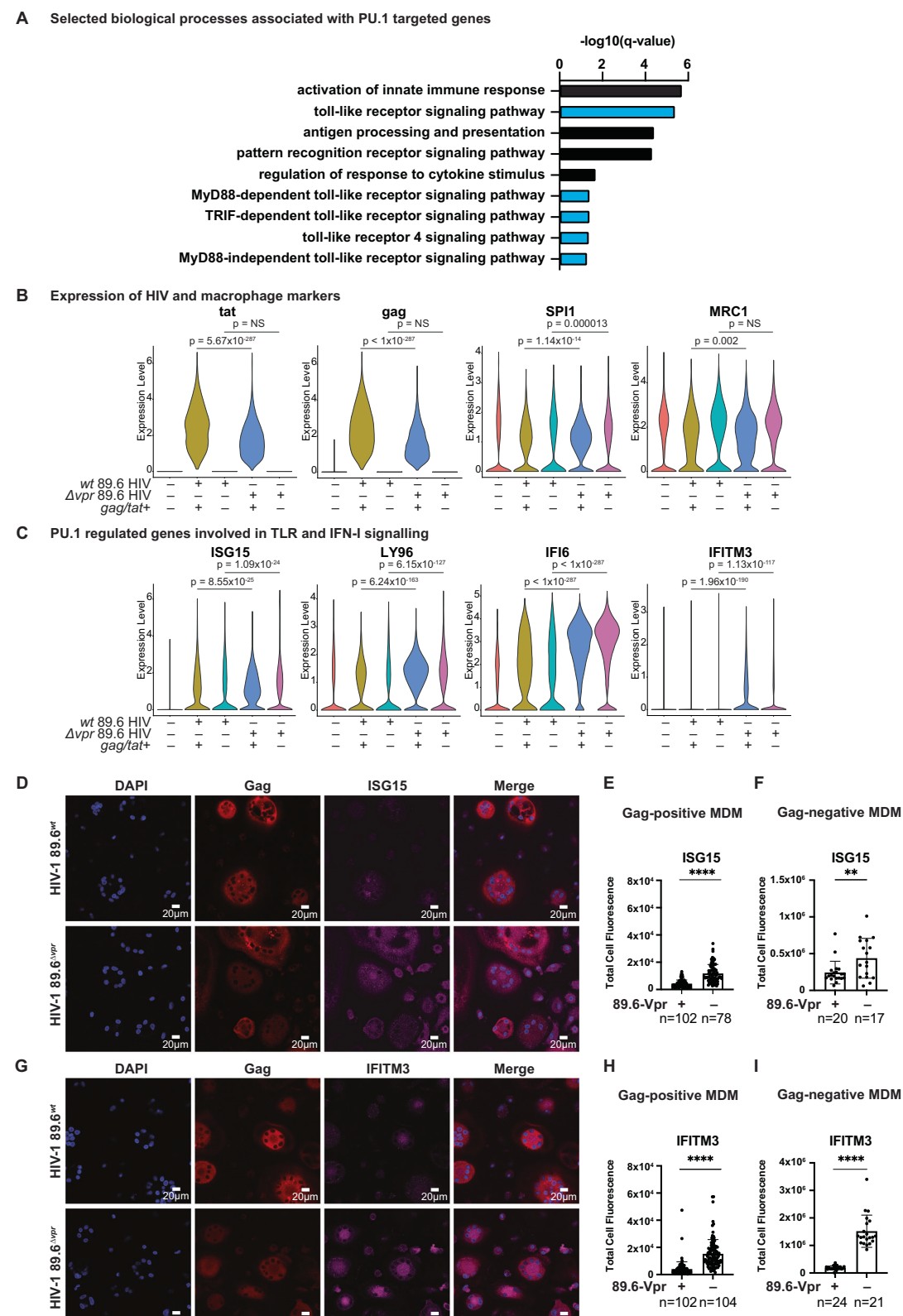

**A** Selected biological processes associated with PU.1 targeted genes

**B** Expression of HIV and macrophage markers

**C** PU.1 regulated genes involved in TLR and IFN-I signalling

**D** DAPI  Gag  ISG15  Merge

**E** Gag-positive MDM — ISG15

**F** Gag-negative MDM — ISG15

**G** DAPI  Gag  IFITM3  Merge

**H** Gag-positive MDM — IFITM3

**I** Gag-negative MDM — IFITM3

indirectly on PU.1 RNA through PU.1 binding sites in the *SPI1* promoter[52].

Prior reports indicated that PU.1 suppresses HIV gene expression in a Tat-reversible manner[53]. If true, Vpr-targeting of PU.1 could explain the upregulation of HIV genes observed in Vpr-expressing HIV-infected macrophages (Fig. 1I). To attempt to confirm these results, we co-transfected HEK 293T cells with a PU.1 expression plasmid (Fig. 4E) and

an 89.6-derived HIV genome that expresses GFP constitutively and mCherry upon HIV activation, allowing us to distinguish transfected cells through GFP expression and HIV-LTR activity through mCherry expression (Supplementary Fig. 4A). To eliminate any complications from Vpr-mediated reduction of PU.1, we used an HIV genome that did not express *vpr*. Transfection of increasing amounts of PU.1 plasmid with the same amount of HIV in all conditions resulted in increasing

**Fig. 3 | Vpr counteracts the innate immune response to HIV infection.**
**A** Selected biological processes associated with infection and inflammation (selected from GO terms represented in Fig. 2D). Pathways were identified as associated with PU.1 regulated genes downmodulated in MDMs infected with 89.6$^{wt}$ or 89.6$^{\Delta vpr}$ virus as determined by expression of *gag* and *tat* transcripts (highlighted in Fig. 2C). The −log10 FDR-adjusted *p*-values (*q*-values) are plotted for each gene ontology term. Blue bars represent terms associated with TLR signaling; Black bars represent related gene ontology terms that are similar, but not directly associated with TLR signaling. **B** Violin plots summarizing single-cell RNA transcripts expressed by MDMs treated with the indicated virus and cultured for 10 days. *Gag + /tat+* are the subset of cells expressing HIV genes within in each culture. *SPI1* is the gene that codes for PU.1. *MRC1* is the gene that codes for mannose receptor. **C** Violin plots summarizing single-cell RNA transcripts expressed by primary human macrophages as in **B**. False-discovery rate corrected two-sided Wilcoxon rank-sum *p*-values are shown above the conditions being compared. **D**, **G** Representative immunofluorescent images of MDMs from a single donor infected with either 89.6$^{wt}$ or 89.6$^{\Delta vpr}$ (MDMs from *n* = 2 independent donors). HIV-infected cells identified by Gag staining. **E**, **H** Quantification of ISG15- or IFITM3-corrected total cell fluorescence in Gag+ cells, or **F**, **I** Gag- cells divided by the number of nuclei in the cell area. The number of cells quantified for each condition is indicated. Error bars represent standard error of the mean, *n* = the number of cells quantified. P values were determined using an unpaired two-sided *t* test. \*\**p* = 0.0096; \*\*\*\**p* < 0.0001. Source data are provided as a Source Data file.

amounts of PU.1 protein as measured by flow cytometry (Supplementary Fig. 4B, C). However, we failed to confirm a PU.1-dependent suppression of HIV-LTR activity (Supplementary Fig. 4D).

## PU.1 downmodulation is a conserved activity of Vprs from HIV-2 and closely related SIV molecular clones

Vpr is highly conserved in lentiviruses, including HIV-2 and all SIV strains[54,55]. Interestingly, HIV-2 and certain SIVs contain both Vpr and Vpx, whereas HIV-1 and some SIVs contain only Vpr (Fig. 5A). The extent to which these two proteins harbor overlapping functions has been the subject of a number of research studies[18,56,57]. Thus, we first examined whether Vprs from viruses containing both Vpr and Vpx would reduce PU.1 levels. To do this, we used the HIV-2$_{ROD}$ molecular clone from which our Vpx control is derived. Transient transfection of HEK 293T cells with a PU.1 expression vector and either HIV-2$_{ROD}$Vpx, HIV-2$_{ROD}$Vpr, or HIV-1$_{89.6}$Vpr confirmed a significant decrease in PU.1 for cells expressing Vpr$^{ROD}$ or Vpr$^{89.6}$ but not Vpx$^{ROD}$ (Fig. 5B, C). This indicates that Vpx and Vpr from the same molecular clone have divergent functions with respect to PU.1 downmodulation.

We next tested the ability of Vpr from six isolates evolutionarily similar or dissimilar to HIV-1 to determine whether the PU.1 targeting function of Vpr is evolutionarily conserved. Vpr sequences from the indicated SIV molecular clones from separate clades, including two from chimpanzee and one from gorilla (the direct evolutionary relatives of HIV-1), were indeed able to mediate the degradation of PU.1 in HEK 293T cells (Fig. 5D, E). Thus, Vpr-mediated reduction of PU.1 was consistently observed across all HIV-1, HIV-2, and SIV molecular clones tested, indicating a strong selective pressure for HIV-related viruses to downmodulate PU.1 in infected cells.

## Both PU.1 and DCAF1 form a complex with Vpr

To determine whether Vpr recruits PU.1 similarly to other host proteins for proteasomal degradation via the CRL4-DCAF1 ubiquitin ligase complex (reviewed in ref. 54), we first assessed whether Vpr and PU.1 formed a complex in transfected HEK 293T cells. We found that PU.1 efficiently co-precipitated with FLAG-tagged 89.6-Vpr. In addition, and as expected[16,17], DCAF1 also co-precipitated with Vpr (Fig. 6A).

We next examined whether Vpr co-precipitated with endogenous PU.1 in the K562 cell line (originally derived from a patient with CML)[58]. To accomplish this, we stably expressed FLAG-tagged HIV-1$_{89.6}$Vpr or HIV-2$_{ROD}$Vpx in K562 cells using lentiviral vectors and repeated co-immunoprecipitation experiments. We found that both PU.1 and DCAF1 co-precipitated with Vpr to a greater extent than Vpx, again suggesting PU.1 binding is specific to Vpr and not to the similar accessory protein Vpx (Fig. 6B). Finally, to determine if Vpr and DCAF1 interact with PU.1 in macrophages, we expressed FLAG-tagged HIV-1$_{89.6}$Vpr in MDMs using a lentiviral vector. As shown in Fig. 6C, we determined that both endogenous PU.1 and DCAF1 interact with Vpr in this physiologically relevant target of HIV-1. Collectively, these data indicate that Vpr selectively forms complexes with both PU.1 and DCAF1 in cells endogenously expressing PU.1.

## Interactions between PU.1 and Vpr require DCAF1

To better understand the mechanism through which Vpr downmodulates PU.1, we assessed a potential requirement for the host protein DCAF1. DCAF1 (also known as VprBP[16]) is the Vpr binding partner in the CRL4 ubiquitin ligase complex that is necessary for proteasomal degradation of Vpr-recruited host proteins[10,11,14,17,18]. DCAF1 requirement was tested using a K562 cell line expressing a lentiviral vector containing either a non-targeting shRNA (shScramble) or a DCAF1-targeting shRNA cassette (shDCAF1). We found that the amount of PU.1 that co-precipitated with Vpr was decreased with DCAF1 silencing (Fig. 6D), suggesting a role for DCAF1 in the interaction between Vpr and PU.1. Consistent with this, endogenously expressed PU.1 from K562 cells did not coprecipitate with a Vpr mutant (89.6-Vpr$^{Q65R}$) protein that is defective for DCAF1 interactions[59,60] (Fig. 6E). These results suggest the unexpected conclusion that PU.1, Vpr, and DCAF1 may form a trimeric complex and PU.1 binding in the complex is dependent on both Vpr and DCAF1, though Vpr can associate with DCAF1 independently of PU.1.

To validate this physical interaction, we performed the immuno-precipitation (IP) in reverse using HEK 293T cells that overexpressed FLAG-tagged-PU.1. Because these experiments used un-tagged Vpr, they were limited by the availability of antibodies, which do not recognize all Vpr allotypes equally. To overcome this limitation, we performed FLAG-tagged PU.1 pull-down experiments with NL4-3 Vpr, which was efficiently recognized by the available antibodies whereas 89.6 Vpr was not. As we observed in K562 cells using 89.6 FLAG-tagged Vpr, the amount of NL4-3 Vpr that co-precipitated with FLAG-tagged PU.1 in HEK 293T cells was decreased with DCAF1 silencing (Fig. 6F). In addition, the Vpr mutant that is defective at interactions with DCAF-1 (NL4-3-Vpr$^{Q65R}$) was also defective at co-precipitating PU.1 (Fig. 6F), further confirming a role for DCAF1 in the interaction between Vpr and PU.1.

In comparison to single cell and flow cytometric approaches shown in Figs. 4 and 5, which had the ability to differentiate infected from uninfected cells, western blot analysis of input lysates shown in Fig. 6 had a low sensitivity to detect Vpr-mediated PU.1 degradation. This was likely due to overexpression of tagged exogenous PU.1, and heterogeneous mixtures of transfected cells that were not optimized to ensure that Vpr was expressed in all PU.1 positive cells.

## TET2, a PU.1 cofactor and Vpr target, coprecipitates with PU.1, Vpr, and DCAF1

The PU.1-regulated antiviral factor, *IFITM3* (Fig. 2C, E) is also controlled by TET2, another target of Vpr[38,39]. TET2 is a DNA dioxygenase that demethylates the *IFITM3* promoter during viral infection, inhibiting HIV Env trafficking and reducing viral spread[61,62]. To prevent this, Vpr mediates the ubiquitylation and degradation of TET2, which in turn inhibits *IFITM3* expression. Interestingly, PU.1 and TET2 have been reported to form a complex to co-regulate myeloid-specific genes[25,26,32]. We therefore hypothesized that TET2 could be co-recruited with PU.1 to DCAF1 by Vpr for degradation. Indeed, we

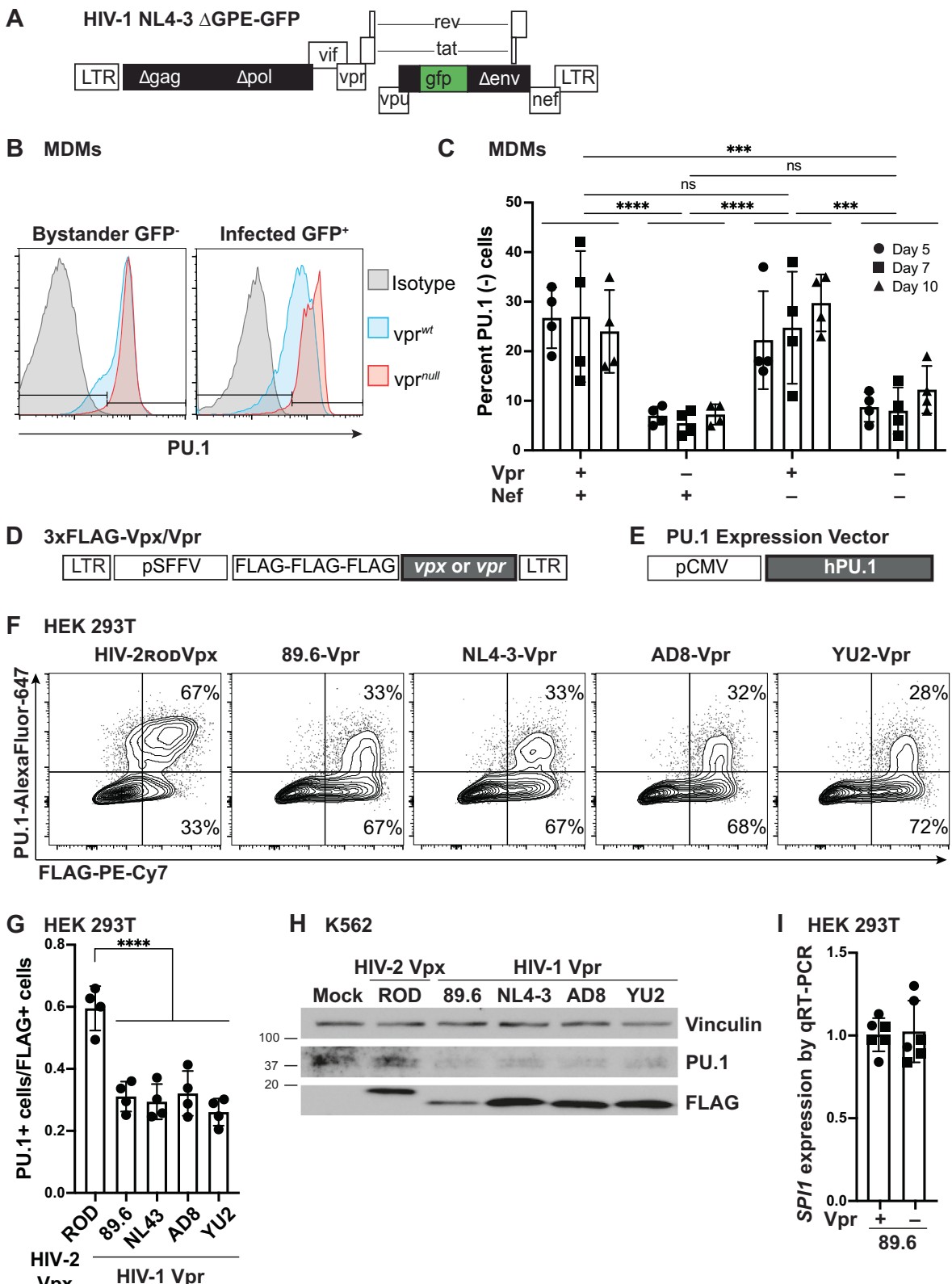

found that TET2 co-precipitated with PU.1 in HEK 293T cells with Vpr and DCAF1 (Fig. 6G).

**Vpr downmodulates PU.1 via a pathway that depends on proteasome activity**

Having shown that Vpr requires DCAF1 to form a stable complex with PU.1, we next asked whether interactions with DCAF1 were also

important for promoting PU.1 degradation. Compared to the flow cytometric approaches shown in Figs. 4 and 5, detection of degradation by western blot is more challenging. The model systems described in Fig. 7 did not reliably show PU.1 degradation, most likely because PU.1 was overexpressed and/or Vpr was not expressed in a sufficient number of PU.1 expressing cells. Therefore, to further study PU.1 degradation, we employed another approach in which virion-

**Fig. 4 | PU.1 levels decrease in the presence of Vpr. A** NL4-3 ΔGPE-GFP viral genome map. **B** Representative flow cytometry histogram of PU.1 expression in infected (GFP⁺) or uninfected bystander (GFP⁻) MDMs infected with NL4-3 ΔGPE-GFP with or without *vpr* and collected on day 7 post infection. **C** Summary graph showing the percentage of infected (GFP⁺) cells that do not express PU.1 as determined by flow cytometry as depicted in Fig. 4B. The mean ±standard deviation from *n* = 4 independent donors is shown for each time point. *P* values were determined using a two-sided, one-way analysis of variance (ANOVA) with Tukey's multiple comparisons test. ***p < 0.001; ****p < 0.0001. **D** Lentiviral map of vectors encoding 3xFLAG-encoding *vpr* or *vpx* genes. **E** PU.1 expression plasmid map for full-length, human PU.1. **F** Representative flow cytometric plots of HEK 293T cells transiently transfected with PU.1 and the indicated FLAG-tagged viral protein.

**G** Summary graph showing PU.1⁺ expression in transfected (FLAG⁺) cells. Each point represents the mean of three technical replicates. The mean ± standard deviation of four independent experiments is shown. *P* values were determined using one-way ANOVA compared to control; ****p < 0.0001. **H** Immunoblot analysis of PU.1 in K562 cells transduced with the indicated lentivirus expressing 3xFLAG-tagged viral proteins. Results are representative of those from three independent experiments. **I** Summary graph of *SPI1* (the gene encoding PU.1) expression in HEK 293T cells co-transfected with PU.1 and an 89.6 expression vector with or without an intact *vpr* open reading frame. *SPI1* levels were assessed from purified RNA using RT quantitative real time PCR. Results represent the mean fold change compared with wild type ±standard deviation for samples performed in triplicate for two independent experiments. Source data are provided as a Source Data file.

associated Vpr is delivered to primary MDMs, resulting in Vpr-dependent proteasomal degradation of targets within five hours of viral treatment[34]. Using this approach, we observed Vpr-dependent degradation of PU.1 treated with wild type HIV-1 but not an HIV harboring a Vpr mutant defective at interacting with DCAF1 (89.6-Vpr^{Q65R}) or 89.6-Vpr-null (Fig. 7A, B). These findings support the conclusion that interactions between PU.1, DCAF1, and Vpr are needed for efficient PU.1 degradation. Additionally, this experiment suggests that it is possible for virion-associated Vpr to act on uninfected bystander macrophages, potentially explaining some of the results from Fig. 3 showing that Vpr can suppress innate immune responses in uninfected bystander cells.

Because the Vpr/DCAF1 complex promotes ubiquitylation and proteasomal degradation of Vpr-bound host proteins, we investigated whether inhibition of the proteosome could restore PU.1 levels. As shown in Fig. 7C, where we optimized PU.1 expression in HEK 293T cells expressing Vpr, PU.1 levels were restored with MG132, a specific inhibitor of the proteasome. Furthermore, MG132 treatment of MDMs transduced with 89.6^{wt} (as for Fig. 7A) prevented virion-associated Vpr-mediated degradation of endogenous PU.1 (Fig. 7D, E).

The results from the immunoprecipitation and degradation assays indicate a requirement for DCAF1 for Vpr to bind PU.1 and promote its degradation, suggesting all three molecules interact as modeled in Fig. 7F. While other explanations remain possible, our data suggest PU.1 binding in the complex is dependent on both Vpr and DCAF1. Vpr can associate with DCAF1 independently of PU.1 and PU.1 can associate with TET2 independently of Vpr.

### Reducing PU.1 enhances HIV-1 Env production in MDMs

Based on scRNA-seq data (Fig. 1), Vpr reduces expression of several antiviral factors regulated by the transcription factor, PU.1 (Fig. 2). Two of these genes, *MRC1* and *IFITM3*, inhibit the spread of HIV in MDMs by targeting HIV-1 Env[35,38]. Therefore, we hypothesized that reduction of PU.1 in MDMs would restore Env production in Vpr-*null*-HIV-infected primary macrophages. To test this hypothesis, we modified the 89.6 genome to remove *vpr* and replace it with either an shScramble sequence, or one of three different shRNAs targeting the PU.1 gene, *SPI1* (Fig. 8A). All three PU.1-targeting cassettes were independently validated in K562 cells to confirm their ability to reduce endogenous PU.1 expression (Fig. 8B). Consistent with our hypothesis, MDMs from two independent donors infected with all three shSPI1-containing viruses showed a marked increase in Env compared to MDMs infected with virus expressing shScramble (Fig. 8C). Overall, our findings support a model in which Vpr dramatically alters the transcriptional landscape in macrophages by targeting myeloid-specific transcription factors that are required for the expression of key antiviral restriction factors, including those that target HIV-1 Env. (Fig. 8D).

### Discussion

Although HIV accessory proteins have been widely studied, the critical function that drives evolutionary conservation of Vpr remains largely enigmatic. While Vpr significantly enhances infection of macrophage-

containing cultures, primarily by enhancing cell-to-cell transmission, it does not substantially affect HIV infection in cultures of CD4⁺ T cells that lack macrophages[1,3–5,33–35]. While a number of Vpr targets implicated in post-replication DNA repair have been identified, there is a lack of compelling explanations for the dramatic selective effects of Vpr on infection and spread in macrophage-containing cultures. In this work, we identified a macrophage-specific target of Vpr, the myeloid transcription factor PU.1 and its associated co-factors that Vpr targets, averting antiviral effects (Fig. 8D). Using single-cell RNA sequencing of MDMs treated with replication-competent virus with and without the gene for Vpr we could distinguish effects of Vpr on cells harboring bona fide infections as well as bystander cells.

In infected cells, we found that Vpr reduces the transcription of hundreds of genes regulated through PU.1 and its cofactors. Several PU.1 regulated genes we identified as being impacted by Vpr are involved in TLR signaling. TLRs are highly conserved PRRs that help cells identify PAMPs and respond quickly to infection. Activation of TLRs through binding of a PAMP, initiates a cascade of intracellular signaling that results in the release of inflammatory cytokines and upregulation of type I interferon response genes[47,49,63]. The products of these genes have antiviral effects on HIV and include *ISG15, STMN1, IFI6, LY96, TREM2, FABP4, IFITM3*, and *MRC1*. *STMN1* is thought to play a role in the establishment of HIV latency, and its depletion leads to higher expression of HIV-1[64]. *MRC1* and *IFITM3* interrupt Env trafficking, reducing viral spread. *IFI6, LY96*, and *ISG15* are members of the type I IFN response to infection[48,65,66], and *ISG15* also inhibits HIV spread by disrupting Gag polymerization[67].

Unexpectedly, we found that bystander cells within HIV-infected cultures also responded to HIV infection by upregulating a subset of these antiviral genes, including *ISG15, LY96, IFI6*, and *IFITM3*. Moreover, we found that upregulation of these factors was reduced in bystander cells from Vpr-positive versus Vpr-negative HIV-infected primary macrophage cultures. Because Vpr is efficiently packaged into the virus particle through specific interactions with Gag p6[2], the Vpr phenotype we observed in bystander cells could be due to a low and/or transient presence of Vpr in MDMs that have taken up viral particles but remained uninfected. This could occur if a subset of Vpr-containing viral particles were defective and/or if innate immune responses blocked completion of reverse transcription and/or integration. We did not consistently detect effects of Vpr in uninfected bystander cells exposed to VSV-G pseudotyped replication-defective virus five days post-infection, but this is not surprising given the likely turnover of the viral protein and the lack of continuous exposure to virions by MDMs treated at a single time point with replication-defective viruses. In contrast, we did find that MDMs exposed to wild type HIV-1 had low PU.1 levels at short time points (five hours) following exposure to virus. Thus, the Vpr-dependent bystander phenotype we identified in MDMs continuously exposed to wild type replication-competent HIV is most likely due to Vpr delivered by wild type viral particles. However alternative and additional explanations are possible. For example, it is possible that there is differential antiviral cytokine production by wild type versus Vpr-mutant infected

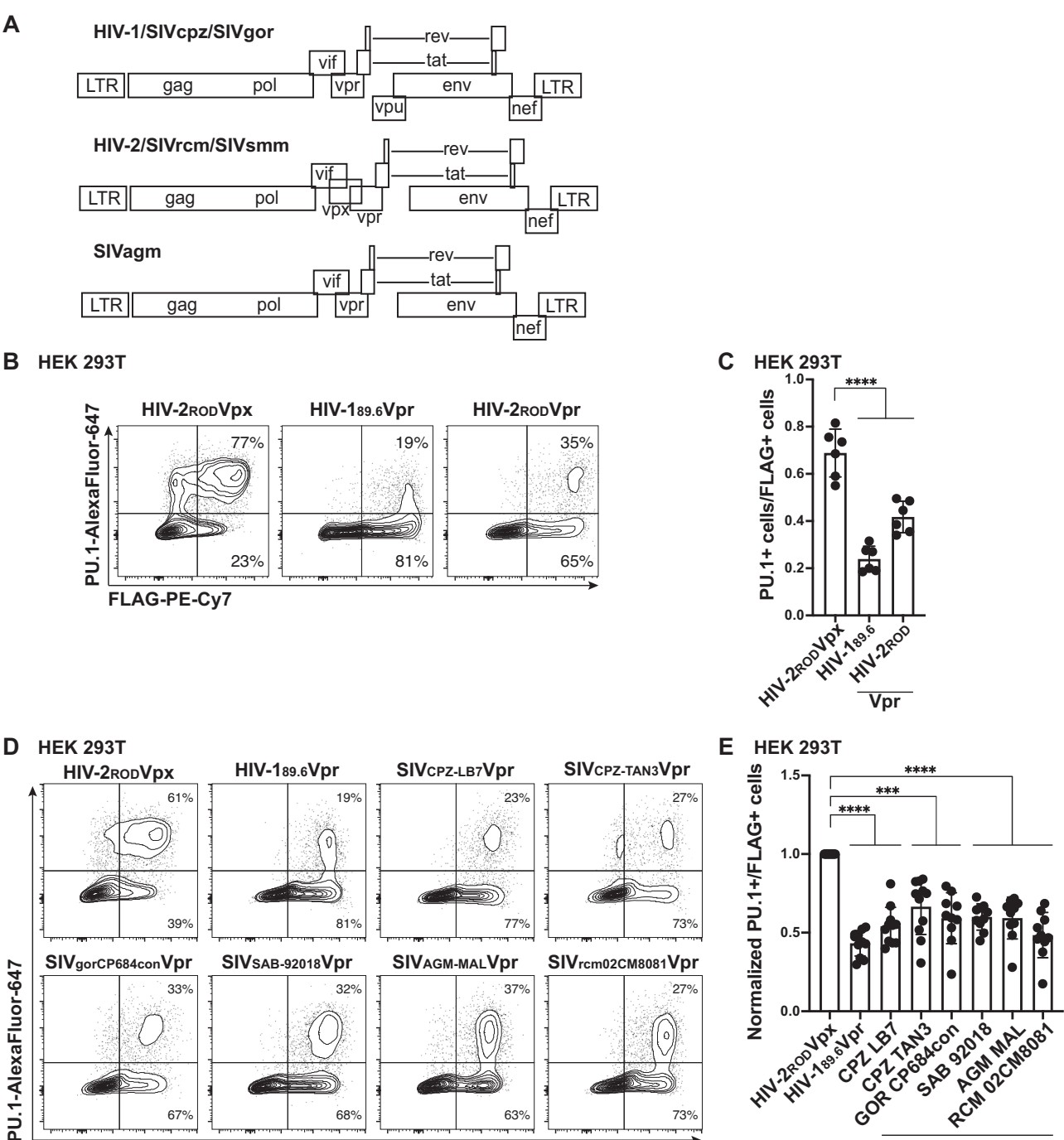

**Fig. 5 | Vpr-mediated reduction of PU.1 is conserved in HIV-2 and SIV molecular clones. A** Genomic maps for HIV-1, HIV-2, and select SIV genomes. HIV-1, SIVcpz, and SIVgor genomes contain *vpr* and *vpu* genes but not *vpx*. HIV-2, SIVrcm, and SIVsmm contain *vpx* and *vpr*. SIVagm contains *vpr*. **B, D** Representative flow cytometric plots of HEK 293T cells transiently transfected with expression plasmids for PU.1 and the indicated FLAG-tagged viral protein. **C, E** Summary graph of data from **B** and **D**, respectively. The percentage of PU.1⁺ cells per transfected (FLAG⁺) cells is shown. Each point represents the average of three technical replicates. The mean ± standard deviation is shown for $n = 6$ (**C**) or $n = 11$ (**E**) independent experiments, respectively. Part **E** was additionally normalized to HIV-2$_{ROD}$Vpx for each experiment. ****$p < 0.0001$ using two-sided one-way ANOVA compared to control with Tukey's multiple comparisons test. Source data are provided as a Source Data file.

macrophages that acts on bystander cells. Regardless of the precise mechanism, our results indicate that Vpr can exert systemic antiviral affects that favor virus infection and spread.

A prior study reported scRNA-seq analysis of THP-1 cells transduced with a replication-defective HIV[68]. The authors identified a population of cells with low HIV gene expression they felt was due to the presence of unintegrated pre-integration complexes (PIC) and

noted transcriptomic changes within this population of cells compared to the fully infected population. Our study differed from this prior report in that we utilized wild type HIV-infected primary macrophages and characterized Vpr-dependent transcriptomic changes comparing fully infected and uninfected (HIV-RNA-negative), bystander cells. We did not identify a similar population of cells that expressed low levels of HIV gene products.

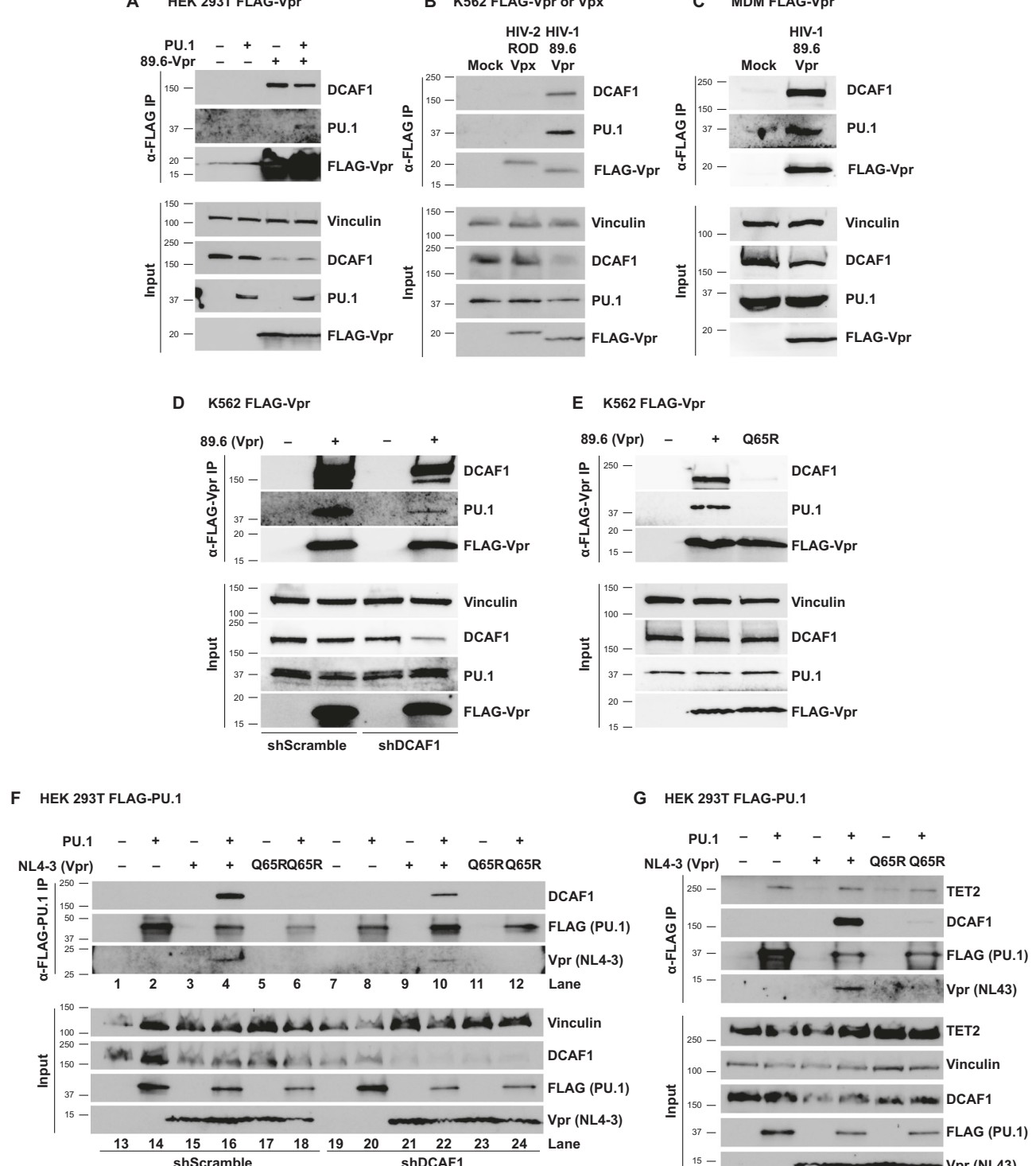

**Fig. 6 | Vpr forms a DCAF-1-dependent complex with PU.1. A** Western blot analysis of lysates from HEK 293T cells co-transfected with the indicated expression constructs and immunoprecipitated with an antibody directed against the FLAG epitope. **B** Western blot analysis of lysates from K562 cells transduced with lentiviruses and treated as in part **A**. **C** Western blot analysis of lysates from MDMs transduced with lentivirus and treated as in part **A**. For parts **A**–**C**, results are representative of three independent experiments. **D** Western blot analysis of lysates from K562 cells stably expressing either control, non-targeting shRNA (shScramble) or an shRNA targeting DCAF1 (shDCAF1), then transduced with FLAG-tagged 89.6-Vpr expression lentivirus and immunoprecipitated using an antibody

directed against the FLAG epitope. **E** Western blot analysis of lysates from K562 cells transduced with FLAG-tagged 89.6-Vpr[WT] or 89.6-Vpr[Q65R] expressing lentiviruses and immunoprecipitated using an antibody directed against the FLAG epitope. **F** Western blot of lysates from HEK 293T cells stably transduced with virus expressing a non-targeting shRNA (shScramble) or one targeting DCAF1 (shDCAF1) and immunoprecipitated with an antibody directed against the FLAG epitope. **G** Western blot analysis of lysates from HEK 293T cells co-transfected with the indicated expression constructs and immunoprecipitated with an antibody directed against the FLAG epitope. Results are representative of three independent experiments. Source data are provided as a Source Data file.

**A** HIV-treated primary macrophages

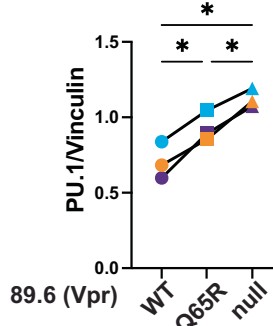

**B** HIV-treated primary macrophages

**C** HEK 293T FLAG-Vpr

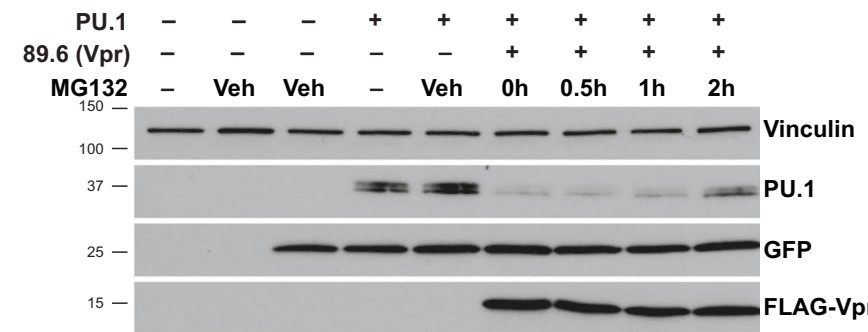

**D** HIV-treated primary macrophages

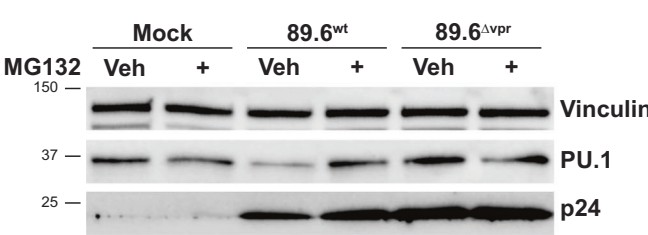

**E** HIV-treated primary macrophages

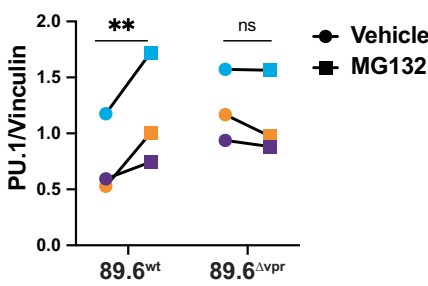

**F** Working model of PU.1 and TET2 interacting with Vpr and DCAF1 in macrophages

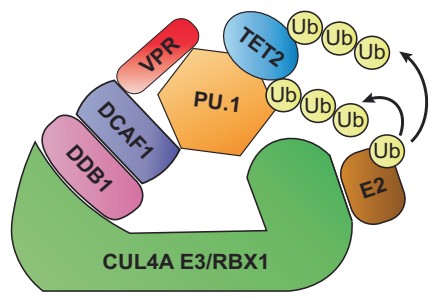

We have previously shown that mannose receptor is a host restriction factor that reduces HIV spread in macrophages by binding to mannose residues on Env and directing Env to the lysosome for degradation[35]. However, our previous data lacked a mechanism for Vpr-mediated transcriptional reduction of *MRC1* in macrophages. The data here confirm findings from other groups that PU.1 regulates *MRC1* expression[44,53], and we provide new evidence that Vpr reduces the transcriptional expression of *MRC1* in macrophages via PU.1 degradation.

In agreement with other studies[38,39], our unbiased survey of Vpr's effect on expression of the host transcriptome showed that Vpr reduced *IFITM3* gene expression substantially. IFITM proteins are broad antiviral factors that inhibit viral entry and exit for HIV-1, SIV, MLV, VSV, EBOV, WNV, among other viruses[69]. Of the IFITM proteins,

**Fig. 7 | An intact DCAF-1 interaction domain is required for Vpr and DCAF1 to degrade PU.1. A** Immunoblot analysis of lysates from MDMs treated for five hours with the indicated virions. **B** Summary graph of PU.1 protein normalized to vinculin from MDMs incubated for five hours with the indicated viruses from **A**. Each point and matched color is representative of an independent donor. Statistical significance was determined using a mixed-effects analysis with Tukey's multiple comparisons test. *$p < 0.05$. **C** Immunoblot analysis of lysates from HEK 293T cells transfected with the indicated expression construct and treatment as indicated with 10 μM MG132 or vehicle (Veh) control (DMSO). A GFP-expressing plasmid was

included where indicated as control for transfection efficiency, $n = 2$.
**D** Immunoblot analysis of lysates from MDMs preincubated for two hours with vehicle (Veh) or MG132 as indicated and then treated for five hours with the indicated virus as in part **A**. **E** Summary graph of PU.1 protein normalized to vinculin from MDMs treated for five hours with the indicated viruses from **D**. Statistical significance was determined using a mixed-effects analysis with Šidák's multiple comparisons test. **$p = 0.0073$. Each point and matched color is representative of an independent donor. **F** Working model of PU.1 and TET2 interacting with Vpr and DCAF1 in macrophages. Source data are provided as a Source Data file.

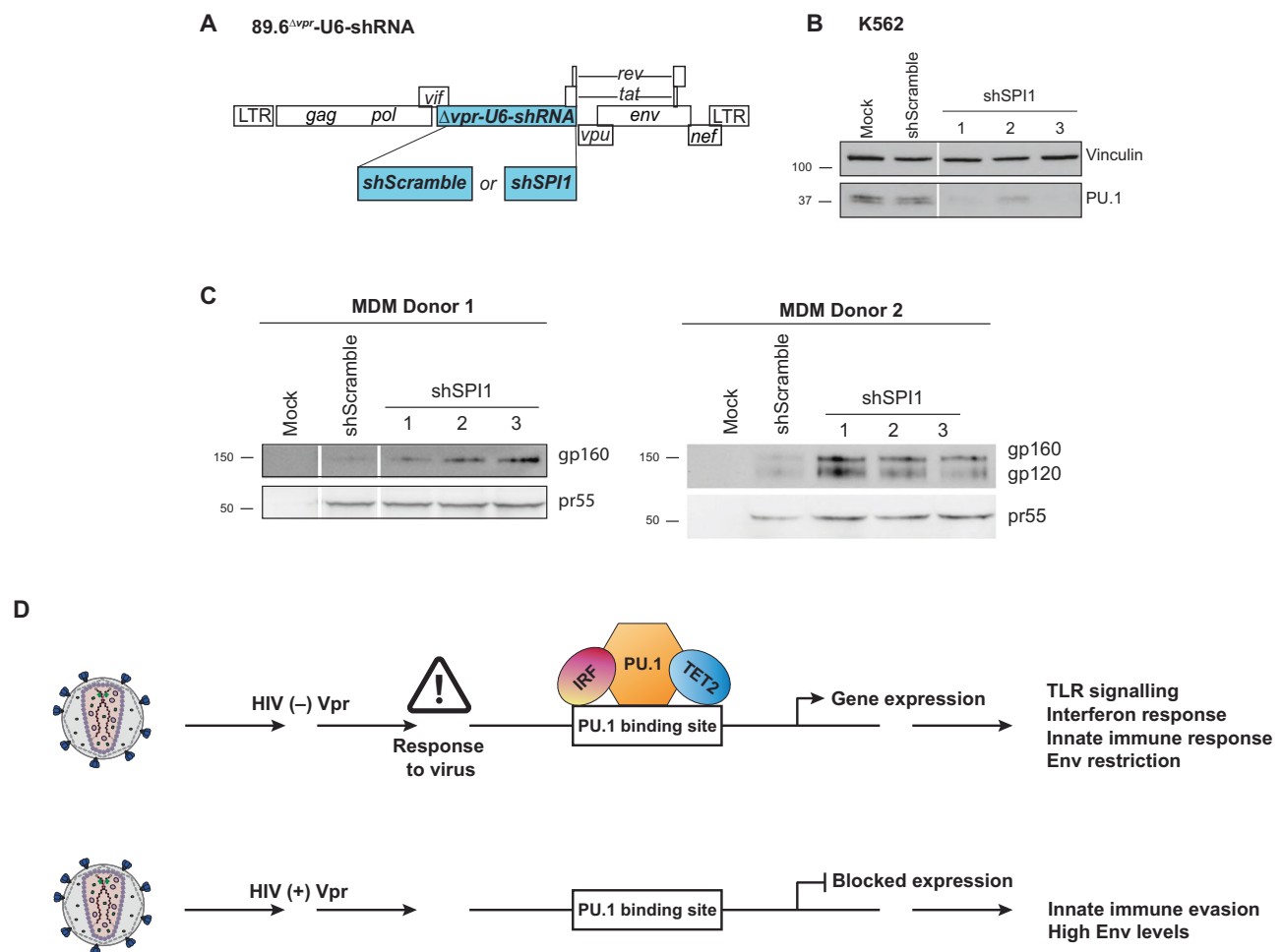

**Fig. 8 | Reducing PU.1 increases Env output in HIV-1 infected MDMs. A** Genome map for full-length 89.6*wt* HIV-1 modified to replace the *vpr* ORF with a U6-promoter followed by either a non-targeting shRNA control (shScramble) or an shRNA targeting the PU.1 transcripts (shSPI1). **B** Immunoblot from K562 cells stably expressing the shRNAs from **A**, $n = 2$. **C** Immunoblot analysis from two independent MDM donors. Lysates collected from MDMs infected with the virus from **A** expressing the indicated shRNA. White lines indicate the location where the digital image of the stained membrane was cropped to remove irrelevant samples. **D** Proposed model

of the PU.1-mediated antiviral response disrupted by HIV-Vpr. HIV (−) Vpr indicates an infection of primary macrophages with HIV that does not express Vpr. PU.1 protein is maintained and available to regulate anti-viral response genes with cofactors such as IRF or TET2. HIV (+) Vpr indicates an infection with HIV that expresses Vpr. PU.1 protein is less available to regulate anti-viral response genes, contributing to innate immune evasion. Source data are provided as a Source Data file.

IFITM3 is well documented as an HIV-1 antagonist. Like mannose receptor, IFITM3 interacts with Env in infected cells, inhibiting Env processing and virion incorporation and strongly inhibiting cell-to-cell spread. Interruption of Env processing by IFITM3 has been demonstrated for several HIV-1 molecular clones[70] many of which were included within our own study (AD8, YU2, and NL4-3, HIV-2$_{ROD}$, SIVagm, and SIVcpz). *IFITM3* expression is regulated by the DNA dioxygenase, TET2[38]. During HIV-1 infection of macrophages, in the absence of Vpr, TET2 demethylates the *IFITM3* promoter, relieving suppression, contributing to the antiviral response. When Vpr is present, *IFITM3* expression is reduced after Vpr recruitment of TET2

to DCAF1 for polyubiquitylation and degradation of TET2[39]. TET2 is ubiquitously expressed in the nuclei of all cells[71]; thus it was unclear how Vpr targeting of TET2 could result in a macrophage-specific Env phenotype. In monocytes, TET2 interacts with PU.1, leading to interaction with genetic targets[32]. Therefore, we hypothesized that Vpr exerts a macrophage-specific effect on the antiviral response by targeting PU.1 and by extension, PU.1-associated proteins. Consistent with this hypothesis, we demonstrated TET2 immunoprecipitating with PU.1, Vpr, and DCAF1 in HEK 293T cells, however confirmation of this interaction in MDMs has not yet been achieved. Altogether, this work provides evidence that Vpr can reprogram cellular

transcription in macrophages by targeting myeloid-specific transcription factors.

A role for PU.1 in Vpr-dependent counteraction of Env restriction was confirmed by replacing the *vpr* ORF with shRNA cassettes targeting *SPI1*/PU.1. This approach allowed us to measure the impact of reducing PU.1 exclusively within HIV-infected macrophages rather than knockdown within the entire culture. This approach was necessary because PU.1 is the master transcriptional regulator necessary for macrophage differentiation and silencing PU.1 prior to infection results in a change in the cellular phenotype that causes resistance to HIV infection[72,73]. While this strategy successfully confirmed that PU.1 knockdown increases Env expression, more research is needed to determine whether other Vpr-dependent transcriptional changes, such as those impacting the PU.1 and interferon-induced gene products ISG15, IFITM3 and IFI6, are mediated through Vpr-dependent degradation of PU.1 alone or whether additional Vpr-dependent pathways play a role.

Vpr-mediated reduction of PU.1 occurred in all cell types tested in our study including MDM, K562, and HEK 293T cells. The Vpr-dependent reduced PU.1 levels were observed regardless of whether PU.1 was expressed from its native promoter or a heterologous promoter. Reversal of PU.1 downmodulation with proteasome inhibitors in both HEK 293T cells and MDMs support the conclusion that Vpr directly reduces PU.1 protein by promoting its degradation. Vpr-mediated degradation of PU.1 was consistent for all HIV-2, SIV, and HIV-1 Group M isolates tested - Group M being largely responsible for the global HIV pandemic. In macrophages, *SPI1* mRNA levels encoding PU.1 protein were also lower in the presence of Vpr, indicating that downmodulation of PU.1 can occur at both the transcriptional and the post-transcriptional level in macrophages.

It is important to note that we failed to confirm prior observations that PU.1 suppresses HIV-LTR activity in a Tat-reversible manner[53]. While it is possible that expression of Tat by our construct prevented our ability to detect an effect of PU.1 on the HIV-1 LTR, these results nevertheless indicate that downmodulation of PU.1 by Vpr is unlikely to impact HIV-1 gene expression in infected macrophages that also express Tat. Thus, in primary HIV-infected macrophages, Vpr mainly targets PU.1 to counteract its anti-viral defense response rather than to counteract an inhibitory effect on HIV gene expression. However, it is possible that the higher expression of HIV genes we observed in Vpr-containing cells may result from Vpr counteracting a PU.1-regulated factor that inhibits HIV gene transcription.

Vpr is highly conserved amongst primate lentiviruses and promotes infection of nondividing cells, especially macrophages[34]. In addition, a requirement for Vpr to achieve maximal replication and persistence in vivo was first discovered using an SIV molecular clone in rhesus monkeys[74]. We therefore speculated that the ability of Vpr to degrade PU.1 is an important evolutionary function of Vpr. HIV-2 differs from HIV-1 in that it contains Vpx, an accessory protein that shares a common genetic ancestor with Vpr[55] but is lacking from all HIV-1 genomes. Thus, we tested the relative ability of Vpx and Vpr from the same molecular clone to promote PU.1 degradation[55]. We identified a unique function of HIV-2 Vpr to degrade PU.1 that was not shared by Vpx. Similarly, both SIVcpz and SIVgor, the evolutionary precursors to HIV-1, and Vpr proteins from more evolutionarily distant viruses consistently lowered PU.1 levels when expressed in the same cell. However, the greatest decrease in PU.1 levels were achieved with HIV-1-derived molecular clones. We therefore speculate that the strength of Vpr-mediated degradation of PU.1 plays an important role in driving spread in human pandemic strains of HIV-1, and PU.1 may be a critical restriction factor, limiting spread between species and within populations. However, more extensive studies comparing the relationship between Vpr and PU.1 across SIV, HIV-2, and HIV-1 isolates is necessary and part of our ongoing efforts.

The relationship between DCAF1 and Vpr is well documented[16–18]. We confirmed the binding of Vpr to DCAF1 and identified PU.1 as a new Vpr-binding factor. We demonstrated that all three components coprecipitate under many different cellular conditions regardless of whether we first precipitated using Vpr or PU.1. To our surprise, mutation of the glutamine residue at position 65 of Vpr (Vpr$^{Q65R}$) not only resulted in the loss of DCAF1 association but also disrupted formation of the Vpr-PU.1 complex. Additionally, when we reduced DCAF1 or used Vpr$^{Q65R}$, the amount of both Vpr and DCAF1 precipitating with PU.1 was reduced. These results indicate that interactions amongst all three proteins are necessary for stable complex formation, although further studies are necessary to understand the detailed protein-protein interactions. Based on these findings, we propose a model in which interactions amongst PU.1, Vpr, and DCAF1 promote the ubiquitylation of PU.1 via the associated CUL4A ubiquitin ligase complex with resultant proteasomal degradation. Consistent with this model, PU.1 was not degraded in the presence of proteasome inhibitors or by a Vpr mutant defective at interacting with DCAF1 in primary MDMs. Thus, our working model is that PU.1 is poly-ubiquitylated following its interaction with Vpr and DCAF1. However complete confirmation of this model has not yet been achieved because we have not yet directly detected ubiquitylated intermediates of PU.1 in Vpr-expressing cells.

Although the role of Vpr in HIV infection has remained largely undefined, we provide evidence that the primary selective pressure for Vpr in lentiviruses is to disrupt the macrophage innate antiviral response to infection, which is achieved by reducing PU.1 levels in infected cells. The ability of Vpr to degrade PU.1 is highly conserved amongst all HIV-1, HIV-2, and SIV isolates tested, and degradation of PU.1 relies on both Vpr and DCAF1. Reducing PU.1 in HIV-infected macrophages lacking Vpr rescued macrophage-dependent restriction of HIV-1 Env, helping to explain the requirement for Vpr in macrophage spreading infections. We are continuing to investigate the transcriptional consequences of PU.1 degradation in macrophages. In addition to TET2, PU.1 associates with other transcription factors, potentially piggybacking other secondary targets to the DCAF1-Cul4A E3 ubiquitin ligase complex via Vpr. Because PU.1 is necessary to maintain macrophage function, future studies should address the greater impact of Vpr-mediated reduction of PU.1 on the infected macrophage.

## Methods
### Ethics statement
Anonymized leukocytes isolated by apheresis were obtained from New York Blood Center after obtaining informed consent. Studies using these cells were determined to be exempt from human studies requirements by the University of Michigan Institutional Review Board because the project involves only biological specimens that cannot be linked to a specific individual by the investigator(s) directly or indirectly through a coding system.

### Cell culture and preparation of human MDMs
All cell cultures were maintained at 37 °C in a 5% $CO_2$ humidified atmosphere. HEK 293Ts (CRL-3216) and K562 (CCL-243) cells were obtained from ATCC and independently authenticated by STR profiling. HEK 293Ts were maintained in DMEM medium (Gibco) supplemented with 100 U/mL penicillin, 100 μg/mL streptomycin, 2 mM glutamine (Pen-Strep-Glutamine, Invitrogen), and 10% fetal bovine serum (Invitrogen). K562 cells were maintained in IMDM (Gibco) and supplemented as HEK 293Ts. To generate monocyte-derived macrophages (MDMs), peripheral blood mononuclear cells were purified by Ficoll density. CD14$^+$ monocytes were positively selected using a CD14$^+$ sort kit following manufacturer instructions (cat# 17858, StemCell Technologies, Vancouver, Canada). CD14$^+$ monocytes were cultured for seven days in R10 [RPMI-1640 with 10% certified endotoxin-low fetal bovine serum (ThermoFisher), penicillin (100 U/mL), streptomycin (100 μg/mL), L-glutamine (292 μg/mL)] supplemented with carrier-free M-CSF and GM-CSF (both at 50 ng/mL, R&D Systems, Minneapolis, Minnesota). Monocytes were plated at $0.5 \times 10^6$ cells/well in a 24-well dish, or $1 \times 10^6$ for lentiviral

transduction with puromycin selection. After seven days the MDMs were treated with virus and maintained in conditioned R10 as described below.

## Viruses, viral vectors, and expression plasmids

The molecular clone p89.6 was obtained from the AIDS reagent program (cat# 3552, from Dr. Ronald G. Collman). The *vpr*-null version was generated as previously described[34]. The human, full-length PU.1 (hPU.1) expression vector was a gift of Dr. Gregory M. K. Poon and generated as previously described[75]. A triple N-FLAG-tagged version of PU.1 was generated by using PCR amplification of hPU.1 to add a KpnI site to the 5′ end and EcoRI to the 3′ end (5′ GTAGGTACCGCCACCATGGAAGGGTT primer and 3′ GTAGAATTCCACCACACTGGACTAGTG primer). The new product replaced an existing gene when inserted into pcDNA3.1 containing a triple N-FLAG-tag between KpnI and EcoRI (Addgene plasmid #67788). The GFP transfection control plasmid, pcDNA3-EGFP, was a gift from Doug Golenbock (Addgene plasmid # 13031; http://n2t.net/addgene:13031; RRID: Addgene_13031). The pNL4-3 ΔGPE-GFP plasmid used for single round infection of macrophages was previously described[76], as were the *vpr*-null, *nef*-null, and *vpr-nef* null versions[35]. pUC19[77] was used as control DNA to adjust transfection samples to the same final DNA concentration. The p89.6-ΔGPEN-mCherry-pSFFV-EGFP single-round infection plasmid was previously described[78] and further modified by replacing *gag* and *pol* with mCherry. Deletion of *vpr* was achieved using Q5 Site-Directed Mutagenesis kit (cat# E0554, New England Biolabs) where the majority of the *vpr* coding sequence was deleted using PCR exclusion (forward - CAGAATTGGGTGTCGACATAG, reverse - TCACAGCTTCATTCTTAAGC). Primers were designed using the NEB Base Changer website (https://nebasechanger.neb.com/) and used following the manufacturer's instructions. pSIV3+ *vpr*-null used in MDM lentiviral transductions to allow Vpx-mediated degradation of SAMHD1 was generated as previously described[35].

Triple FLAG-tagged Vpr and Vpx lentiviral expression vectors for HIV-2$_{ROD}$Vpx (Addgene plasmid #115816), SIV$_{SAB-92018}$Vpr (Addgene plasmid #115822), SIV$_{AGM-MAL}$Vpr (Addgene plasmid #115828), SIV$_{CPZ-TAN3}$Vpr (Addgene plasmid #115833), SIV$_{CPZ-LB7}$Vpr (Addgene plasmid #115834), SIV$_{gorCP684con}$Vpr (Addgene plasmid #115835), and SIV$_{rcm02CM8081}$Vpr (Addgene plasmid #115838) were a gift from Jeremy Luban[57]. Similar vectors for HIV-2$_{ROD}$Vpr, HIV-1$_{89.6}$Vpr, HIV-1$_{89.6}$Vpr$^{Q65R}$, HIV-1$_{NL4-3}$Vpr, HIV-1$_{AD8}$Vpr, HIV-1$_{YU2}$Vpr were generated by synthesizing the gene as a gBlock (IDT, Coralville, Iowa, USA) between NotI and either EcoRI or AflIII in the same lentiviral expression plasmid. Untagged expression vectors for HIV-1$_{NL4-3}$Vpr and HIV-1$_{NL4-3}$Vpr$^{Q65R}$ were generated by synthesizing the genes as a gBlocks (IDT, Coralville, Iowa, USA) and inserting them between SbfI and NotI in LeGO-IV, a gift from Boris Fehse (Addgene plasmid #27360)[79].

The short hairpin RNAs targeting *DCAF1* (target sequence: CCTCCCATTCTTCTGCCTTTA) and *SPI1* (target sequence 1: GCCCTATGACACGGATCTATA, target sequence 2: CGGATCTA-TACCAACGCCAAA, and target sequence 3: CCGTATGTAAATCA-GATCTCC) were designed using Genetic Perturbation Platform (Broad institute) and cloned into pLKO.1 – TRC cloning vector, a gift from David Root (Addgene plasmid # 10878; http://n2t.net/addgene:10878; RRID:Addgene_10878)[80]. The control shRNA, scramble shRNA was a gift from David Sabatini (Addgene plasmid #1864; http://n2t.net/addgene:1864; RRID:Addgene_1864)[81]. For shRNA expression from full-length HIV-1-89.6 virus, the *vpr*-ORF was first disrupted by the insertion of a U6-promoter followed by multiple unique restriction enzyme sequences generated by synthesizing the segment as a gBlock (IDT, Coralville, Iowa, USA). The segment was inserted between XcmI and SalI without disrupting the *vif* or *tat* ORFs. The same shRNAs as above were then cloned into HIV-1-89.6 after the U6-promoter.

## Co-transfections

Co-transfections of 3xFLAG-Vpx/Vpr and hPU.1 were performed in HEK 293T cells. Cells were plated at $1.6 \times 10^5$ per well in a 12-well dish. 24hrs

after plating, 1 ng of hPU.1, 500 ng of 3xFLAG-Vpx/Vpr, and pUC19 to a total of 1 μg of DNA per well were combined with 4 μL of PEI, mixed and added to each well. 48hrs later, cells were harvested for flow cytometry or immunoblotting. For co-immunoprecipitations, transfection experiments were scaled to achieve $60 \times 10^6$ cells per condition. Co-transfections with p89.6-ΔGPERN-mCherry-pSFFV-EGFP and hPU.1 were performed as described above and with DNA amounts described in the legend.

## Transduction of MDM, K562, and HEK 293T

All transductions were performed via spin inoculation at 1050 x g for 2 hr at 25°C with equal virus amounts determined by Gag p24 mass in medium containing 4 μg/mL polybrene (Sigma). MDMs were inoculated with 10 μg p24 mass equivalents of NL4-3 ΔGPE-GFP virus or 20 μg p24 mass equivalents of 3xFLAG-89.6-Vpr. K562 and HEK 293Ts were spin inoculated with 10 μg p24 mass equivalents of shScramble and shDCAF1 viruses. K562 cells were inoculated with varying amounts of 3xFLAG-tagged Vpr/Vpx expression viruses to achieve equal FLAG expression. After infection, viral medium was removed and replaced with fresh medium.

Short hairpin RNA-mediated silencing in MDMs was achieved through spinoculation of freshly isolated primary monocytes with VSV-G-pseudotyped SIV3+ *vpr*-null virus at 1000×g for 1.5 hr with 4 μg/mL polybrene to allow Vpx-mediated degradation of SAMHD1. Cells were then spinoculated with 10 μg p24 mass equivalents of VSV-G-pseudotyped pLKO.1 containing shScramble or shSPI1 lentiviruses at 1000×g for 1.5 h. After virus removal, monocytes were cultured as described above for seven days with R10 containing M-CSF and GM-CSF. At day five, transduced cells were treated with 2.5 μg/mL of puromycin for two days. Thereafter, cells were cultured for an additional 10 days in R10 before harvesting.

## HIV infection of MDM

Prior to infection, half of the medium was removed from each well of MDMs and saved to make diluted conditioned media post-infection. MDM were infected with 5 μg, 10 μg, and 20 μg (scRNA-seq) or 20 μg and 50 μg (immunofluorescence) equivalents of Gag p24 mass diluted in R10 for 6 hr at 37 °C. After the 6 h infection, media was removed and replaced with conditioned media diluted 1:2 in R10. Half-media changes were performed every four days. For assessment of virion-associated impact on PU.1 (including MG132 treatment), MDMs were infected with 200-300 μg of virus in R10 of either 89.6$^{wt}$, 89.6$^{Δvpr}$, or 89.6$^{Δvpr-Q65R}$ for 5hrs.

## Single-cell RNA sequencing

At 10 days post-infection, uninfected, 89.6$^{WT}$, and 89.6$^{Δvpr}$ infected MDMs were lifted using enzyme free cell dissociation buffer (ThermoFisher). Replicate samples were fixed with paraformaldehyde and stained for Gag to assess viral spread by flow cytometry. The resulting flow cytometry data was used to select 89.6$^{WT}$ and 89.6$^{Δvpr}$ conditions with similar percentages of infected cells. Wells of the selected conditions were harvested, counted, and prepared according to manufacturer instructions for 10X Chromium Next GEM Single Cell 3′ v3 Gene Expression (10X Genomics).

## Single-cell data analysis

10X filtered expression matrices were generated from CellRanger version 3.0.0 (10X Genomics). We analyzed all single-cell gene expression data using the standard LIGER[40] (https://github.com/welch-lab/liger) data integration pipeline. All WT and Vpr-*null* infected MDM raw data expression matrices from each donor were combined before merging the data. We used a value of $k = 20$ during joint matrix factorization, resolution of 0.05 for Louvain clustering, and nearest neighbor = 30 with a minimum distance = 0.3 for UMAP visualization. We identified infected cells by sub-setting cells with a non-zero

expression value for both *gag* and *tat* transcripts. We determined differential gene expression between WT and Vpr-*null* infected MDMs using the two-sided Wilcoxin rank-sum test. Volcano plots of differentially expressed genes were generated using ggplot2[82] (https://ggplot2.tidyverse.org/). Downregulated genes in the presence of Vpr with a log2FC > 1 and adjusted *p*-value > 0.05 were used as input for the HOMER[83] (http://homer.ucsd.edu/homer/motif/) 'findMotifs' function using the human reference set. PU.1-motif associated genes were identified using the 'find' function in 'findMotifs' from HOMER. Gene Ontology analysis for biological processes for PU.1 motif-containing genes was determined using GOrilla[84,85] (http://cbl-gorilla.cs.technion.ac.il/) with PU.1 regulated genes as target genes and all expressed genes in our dataset as background. Biological processes were plotted using REVIGO[86] (http://revigo.irb.hr/). The −500bp sequence for MRC1 used for PU.1 motif scanning was obtained from UCSC Genome Browser[87] (https://genome.ucsc.edu/) using Human reference genome GRCh38/hg38. The PU.1 binding motif probability matrix was obtained from HOMER and used with FIMO[88] (https://meme-suite.org/meme/doc/fimo.html) to scan the *MRC1* input sequence. Violin plots were generated by importing our LIGER generated dataset into Seurat[89] (https://satijalab.org/seurat/) and running the VlnPlot function. All single-cell data analysis and plots were done using RStudio[90] (http://www.rstudio.com/) except for HOMER-predicted motifs.

## Virus production

Virus stocks were produced by transfected HEK 293T cells (ATCC, Manassas, Virginia) with viral DNA and polyethylenimine (PEI) (Polysciences, Warrington, PA) as previously described[35]. For replication defective constructs, cells were plated 24hrs before transfection with a DNA ratio of 1:1:1 with pCMV-HIV-1[91], pHCMV-V (VSV-G expression plasmid) (from Nancy Hopkins, Massachusetts Institute of Technology), and lentiviral expression plasmid. Viral supernatant was collected two days post-transfection and stored at −80°C. For infectious virus, pCMV-HIV and pHCMV-V were omitted.

## Virion quantification

Supernatants containing viral particles were lysed in lysis buffer (0.05% Tween 20, 0.5% Triton X, 0.5% casein in PBS). Gag p24 antibody (1 μg/mL, clone 183-H12-5C, cat# 1519 AIDS Reagent Program from Dr. Bruce Cheseboro and Dr. Hardy Chen) was bound to Nunc MaxiSorp plates (cat# 12-565-135, ThermoFisher) at 4°C overnight. Lysed samples were captured at 4°C overnight and then incubated with biotinylated antibody to Gag p24 (1:4000, clone 31-90-25, cat# HB-9725, ATCC) for 1 hr. Clone 31-90-25 was biotinylated with the EZ-Link Micro Sulfo-NHS-Biotinylation Kit (cat# PI-21925 ThermoFisher). Clones 31-90-25 and 182-H12-5C were purified using Protein G columns (cat# 45-000-054, GE Healthcare) following the manufacturer's instructions. Samples were detected using streptavidin-HRP for 30 min (1:10000, Fitzgerald, Acton, Massachusetts) and 3,3′,5,5′-Tetramethylbenzidine substrate (cat# T8665-IL Sigma). Reactions were quenched with 0.5 M $H_2SO_4$. Absorbance was measured at 450 nm with a reference wavelength of 650 nm. CAp24 concentrations were measured by comparison to recombinant CAp24 standards (cat# 00177 V, ViroGen, Watertown, Massachusetts).

## Immunoblots

For western blots, cells were lysed in Blue Loading Buffer (cat# 7722, Cell Signaling Technology), sonicated with a Misonix sonicator (Qsonica, LLC. Newtown, CT), boiled for 10 min at 95°C before loading, and analyzed by SDS-PAGE immunoblot. All uncropped blots can be found in Supplementary Data.

For coimmunoprecipitation, cells were lysed in Pierce IP Lysis Buffer (Thermo Fisher Scientific) and 1x Halt Protease Inhibitor Cocktail (Thermo Fisher Scientific). Lysates were incubated with Anti-FLAG M2 Magnetic Beads (MilliporeSigma, Darmstadt, Germany) according to the manufacturer's instructions. Proteins were eluted using 3xFLAG

peptide (MilliporeSigma, Darmstadt, Germany) and analyzed by SDS-PAGE immunoblot. FLAG-tagged proteins were visualized using Pierce ECL (Thermo Scientific) after treatment with an HRP-conjugated primary antibody directed against the FLAG epitope (Millipore Sigma). HRP-conjugated secondary antibodies against murine and rabbit antibodies to other targets (see below) plus ECL Prime reagent (Cytiva Amersham) were used to visualize all other proteins.

## Antibodies

Antibodies to Vinculin (1:1000, cat# V9131, Millipore Sigma), DCAF1 (1:1000, cat# 11612-1AP, Proteintech), PU.1 (1:100, cat# 2266 S, Cell Signalling Technology), FLAG (1:1000, cat# F1804, Millipore Sigma), TET2 (1:250, cat#MABE462, EMD Millipore), Vpr (1:500, AIDS Reagent Program cat# ARP-11836 from Dr. Jeffrey Kopp), pr55 and p24 (1:1000, AIDS Reagent Program cat# ARP-3957), gp120 (1:1000, AIDS Reagent Program cat# 288), and GFP (1:1000, cat# ab13970, Abcam) were used for immunoblot analysis. Secondary HRP conjugated antibodies against murine (1:10000, rat anti-mouse IgG1, eBioscience), rabbit (1:5000, goat anti-rabbit IgG, cat# 65-6120, Invitrogen), sheep (1:20000, rabbit anti-sheep IgG, Dako), and human (1:10000, goat anti-human IgG, cat# 62-8420, ThermoFisher) were also used. Antibodies to PU.1 (1:100, clone 7C6B05, BioLegend), FLAG (1:3000, cat# 637324, BioLegend) were used for flow cytometry. Antibodies to ISG15 (cat# 15981-1-AP, Proteintech), IFITM3 (cat# 11714-1-AP, Proteintech), and AlexaFluor 647 (A21244, Fisher Scientific) secondary antibody were used for immunofluorescence. Dilutions listed below. CAp24 (1:400, clone KC57-PE cat# 6604667, Beckman Coulter) was used for both flow cytometry and immunofluorescence.

## Immunofluorescence

MDMs were generated as described above in μ-slide glass-bottomed cell chambers (Ibidi, Gräfelfing Germany) and infected as described above. Cells were fixed by adding 4% paraformaldehyde (PFA) and permeabilized by adding 0.1% TritonX-100 in PBS. Cells were then blocked by incubating with 5% goat serum (Millipore Sigma) and 1% bovine serum albumin (BSA) in PBS for 30 mins at room temperature. Primary antibodies against ISG15 (Proteintech) or IFITM3 (Proteintech) were diluted 1:450 or 1:400 respectively in 1% BSA in PBS and were incubated with cells for 90 min at room temperature. Goat anti-rabbit AlexaFluor 647 secondary antibody (Fisher Scientific) was diluted 1:200 in 1% BSA and incubated with the cells for 30 min at room temperature, protected from light. Cells were incubated with Anti-PE conjugated Gag (1:400) antibody for 30 min at room temperature. Cells were washed three times with PBS after each step. Nuclei were stained by diluting a 1 mg/mL stock of DAPI (ThermoFisher Scientific) 1:1,000 in PBS and incubating with the cells for 5 minutes at room temperature. Cells were imaged with a Nikon N-SIM + A1R confocal microscope. Identical laser and gain settings were used across all images for each individual replicate of the experiment. Images were processed using NIS viewer imaging software and corrected total cell fluorescence (CTCF) was calculated using Image J software[92]. CTCF = Integrated Density − (area of selected cell x mean fluorescence of background readings). Total corrected fluorescence per cell was divided by the number of nuclei to normalize cell volume to account for multinucleated syncytia.

## Quantitative RT PCR

HEK 293 T cells sorted as described in 'Flow Cytometry' below were counted using the Countess II Cell Counter (Thermo Fisher Scientific), and cell samples were diluted such that all conditions contained the same cell numbers as input. RNA was isolated using the Zymo Direct-Zol RNA MiniPrep Plus extraction kit with an on-column DNaseI digestion. RNA was reverse transcribed using the High-Capacity cDNA Reverse Transcriptase kit (Applied Biosystems). Quantitative PCR was performed using SYBR green qPCR Master Mix (Applied Biosystems) on a QuantStudio 3 Real-Time PCR System (Applied Biosystems) with

ReadyMade PrimeTime primers for *SPI1* (cat# Hs.PT.58.19735554, Integrated DNA Technologies Inc, USA) and RT2 qPCR Primer Assay for Human GAPDH (cat# PPH00150F-200, Qiagen). Expression was quantified using ABI Sequence Detection software compared to serial dilutions of an SPI1 or GAPDH synthetic sequence gBlock (Integrated DNA Technologies Inc, USA). Measured values for *SPI1* were normalized to measured values of *GAPDH*.

## Flow cytometry

For cells requiring intracellular staining using antibodies directed against HIV Gag p24, FLAG-Vpx and -Vpr, and PU.1, paraformaldehyde-fixed cells were permeabilized with 0.1% Triton-X100 in PBS for 2 min followed by incubation with antibody for 30 min at room temperature. In all experiments, cells were gated sequentially by forward scatter vs. side scatter for cells and then by forward scatter area vs. height to exclude doublets. The gating strategy is shown in Supplementary Fig. 3B. All transduced MDMs and transiently transfected HEK 293T cells were assessed for protein expression on the Cytek Aurora. GFP+ HEK 293T cells in the MG132 treatment experiments and for RT-qPCR were sorted on the Sony SH800 cell sorter into R10. Untreated, GFP- cells were also sorted. All flow cytometry data was analyzed using FlowJo v10 software (BD Life Sciences).

## Proteasome inhibition

Lyophilized MG132 was purchased from MilliporeSigma (cat# M8699) and dissolved in DMSO. For HEK 293Ts, MG132 was added to cellular medium to achieve a final concentration of 10 μM at varying timepoints. Cells were harvested from replicate wells and all drug treatment timepoints were collected at once. DMSO-only control treatment wells (Vehicle) were treated with the same volume of DMSO as contained in the MG132 treatment conditions. For MDMs, cells were pretreated with 2.5 μM MG132 for 2hrs prior to infection, then maintained in 2.5 μM MG132 throughout the infection.

## Statistical analysis

All non-single cell statistical analyses were performed using GraphPad Prism v10 Software (Boston, MA) as described in figure legends for each experiment.

## Reporting summary

Further information on research design is available in the Nature Portfolio Reporting Summary linked to this article.

## Data availability

CellRanger version 3.0.0-processed data generated in the manuscript have been deposited in GEO under accession code GSE220574. Raw sequencing data have been deposited in NCBI dbGAP database under accession code phs002915.v2.p1. The raw data are available under restricted access due to data privacy concerns and can be obtained by requesting access from NCBI. TF motif data and analysis from HOMER[83] (http://homer.ucsd.edu/homer/) are described above. Source data are provided as a Source Data file. Source data are provided with this paper.

## Code availability

We analyzed all single-cell gene expression data using the standard LIGER[40] (https://github.com/welch-lab/liger) multiple single-cell RNA-seq data integration pipeline. TF binding motif data was generated using the HOMER[83] (http://homer.ucsd.edu/homer/motif/) 'findMotifs' function using the human reference set. PU.1-motif associated genes were identified using the 'find' function in 'findMotifs' from HOMER.

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

## Acknowledgements

This work was supported by National Institutes of Health grants R01AI149669 and 1R61DA059916-01 to K.L.C. and J.D.W., F31AI15504 to M.C.V. and F31AI125090-01 to J.L., R21AI32379 to K.L.C, training grants T32GM007315 to M.C.V., and 5T32GM008353-27 to J.L. Additional funding provided by the Rackham Regents Fellowship to M.C.V, and The Barry Goldwater Scholarship, The University of Michigan LSA Honors Summer Fellowship, and Otto Graf Scholarship to T.C. We thank Chen Li from the Welch lab for his invaluable advice and aid with single-cell data analysis. We thank the University of Michigan Flow Cytometry Core and Advanced Genomics Core for assistance with experiments, particularly Wenpu Trim and Tricia Tamsen. Thank you to all members of the Collins lab for their helpful discussions, particularly Valeri Terry for her assistance with cloning and Francisco Gomez-Rivera for his assistance with RT-qPCR.

## Author contributions

M.C.V., J.L., and K.L.C. conceptualized the study; M.C.V, J.L, J.D.W., and K.L.C. designed the methodology; software usage determined and executed by M.C.V. and J.D.W.; M.C.V. performed validation and formal analysis of all experimentation with help from B.R.; Investigation performed by M.C.V., B.R., T.C., W.M.D., and J.L.; Resources provided by K.L.C.; M.C.V. wrote the manuscript, with review, editing, and input from J.D.W., and K.L.C.; Visualization of data was done by M.C.V.; All authors commented on the manuscript. All contributions were supervised by J.D.W. and K.L.C.; Funding was acquired by M.C.V., J.D.W., and K.L.C.

## Competing interests

The authors declare no competing interests.
