## [Peer Review File · Nature Communications]

HIV-1 Vpr combats the PU.1-driven antiviral response in primary human macrophagesREVIEWER COMMENTS

Reviewer #1 (Remarks to the Author):

"Vpr, is an enigmatic protein required for efficient spread of HIV from macrophages to T cells". VPR's pleiotropic effects have been hard to nail down because it is widely accepted that they are mediated via a pleiotropic mechanism of ubiquitination and proteasomal degradation targeting various selected proteins. This study presents and examines the significant and noteworthy finding that PU.1 is a target of VPR mediated degradation. PU.1 is a myeloid specific transcription factor that is required for efficient induction of the host innate immune response to HIV. By degradation of PU.1 VPR can effectively attenuate transcription of many innate response transcripts and mute the host's defensive response to HIV infection.

scRNA seq and bioinformatic analyses suggest the hypothesis that PU.1 is a key target of VPR degradation. This work tests that notion using a number of HIV and other lentiviral analog constructs with contrasting WT and inactive versions of VPR (and analogs). Flow cytometry, Western Blot and pull down experiments quantify protein levels and interactions. The work meets the new standard for the field in which omics observations are "ground truthed". Experimentally, it is well designed and executed, yielding convincingly interpretable data. The methodology is sound and the information detail adequate for repetition.

As always, good work leaves the reader wanting more: While the whole premise of the proposed mechanism is that the VPR-mediated proteasome degradation of PU.1 and TET2 results in the down regulation of macrophage innate response gene transcripts, there is very little investigation/quantification of the ubiquitination of these targets or the specific proteins involved in their degradation. While convincing evidence of the VPR dependency of reduced levels of these key proteins is presented (closely linked to transcriptomic quantification of responsive genes), the work stops there.

Only a few typos were noted. It is recommended that this manuscript be accepted for publication.

Reviewer #2 (Remarks to the Author):

The manuscript by Virgillo et al studies the role of HIV-1 Vpr in monocytes by examining the impact on cellular transcription both in infected cells and in bystander cells within the infected pool. They center their analysis on the downregulation of transcription factor PU.1 which alters upregulation of innate immune genes ISG15, LY96 and IFI6. The authors suggest a model whereby Vpr inhibits the induction of innate antiviral immune programs by binding to and inducing DCAF-1 dependent degradation of PU.1. The hypothesis presents a novel mechanism and explains a dominant transcriptional changes that occur only in bystander cells, not in the infected cells. Overexpression and knockdown experiments to further test their hypothesis are well performed, and evidence for a direct interaction with the PU.1 transcription factor is convincing with co-IP studies. The study supports a novel mechanism of action of Vpr that can prevent interferon responses in infected cells, by directly downregulating a key required transcription factor, and shows that this is a conserved activity of many alleles of vpr from SIV, HIV-2.

There appears to be a modest benefit to the WT Vpr expressing infected cells which have higher HIV gene expression. Nonetheless, the overall phenotype regarding how this impacts viral replication or persistence is still unclear. The discussion may benefit from some additional attention to how the authors believe this activity benefits the virus as the role of Vpr remains somewhat enigmatic.

Minor note:

Figure 4B. In the bystander GFP- cells there appears to be some downregulation of PU.1 as well when compared to the vpr null. Is this evidence of downregulation of PU.1 in cells that may not yet be expressing HIV, perhaps through virus particle associated vpr? Analysis of the flow plots with 2 colors GFP vs PU.1 may be especially informative here.

Reviewer #3 (Remarks to the Author):

In this study, Virgilio et al., used single cell RNAseq to capture the transcriptional changes mediated by Vpr during HIV-1-infection of macrophages. The authors previously demonstrated that Vpr counteracts accelerated degradation of the Env glycoprotein initiated by the mannose receptor (MR) by suppressing expression of the MR gene, MRC1. However, the mechanism by which Vpr suppresses MRC1 expression wasn't known. Here they show that Vpr reduces the levels of PU.1, a transcription factor regulating many essential macrophage genes, including MRC1. These timely results suggest a model in which Vpr could reprogram HIV-1-infected macrophages by targeting PU.1. Accordingly, Vpr was shown to disrupt the expression of several PU.1 associated antiviral factors, including MRC1, but also IFITM3, ISG15, IFI6 and LY96. Importantly, the capacity of Vpr to reduce PU.1 levels was conserved among HIV-1, HIV-2 and SIV. While data presented suggest that PU.1 and DCAF1 form a complex with Vpr, it remains unclear whether PU.1 degradation requires direct Vpr-PU.1 interaction and the recruitment of the DCAF1-Cul4 E3 ubiquitin ligase complex. Moreover, direct experiments connecting Vpr-mediated PU.1 degradation to its capacity to regulate gene expression and favor HIV replication in MDMs are needed.

Comments

1- In Figure 2E and 3C, the authors show that Vpr reduces the expression of several transcripts involved in TLR and IFN-I signaling, such as ISG15, IFITM3 and IFI6 that contribute to host innate immune response to HIV. How this impacts the level of the proteins encoded by these transcripts was not reported. Evaluation of the levels of the proteins encoded by these transcripts (in HIV-1 infected and uninfected bystander MDMs) would confirm the impact of Vpr on host innate immune response.

2- The author's main conclusions are that Vpr reprograms HIV-1-infected MDMs gene expression by targeting PU.1 and that contributes to disrupt expression of several antiviral factors (MCR1, ISG15, IFITM3, LY96 and IFI6) restricting HIV infection and spread. Evaluating the impact of PU.1 depletion on the capacity of Vpr to modulate antiviral gene expression in MDMs therefore favoring HIV replication would strengthen the conclusions of the manuscript.

3- The authors previously demonstrated that Vpr counteracts accelerated degradation of Env by reducing MCR1 expression. It is unclear why the authors didn't evaluate the impact of PU.1 depletion on Vpr's ability to reduce Env expression. Surprisingly, the only experiment evaluating the impact of Vpr on PU.1 levels in MDMs was performed using an Env-defective virus (Figure 4A,B,C). This looks like a straightforward experiment for this group and it would greatly benefit the manuscript.

4- SPI1 mRNA levels, encoding for PU.1 were also lower in the presence of Vpr (Figure 2B and 3B), suggesting that PU.1 downmodulation could occur at the transcriptional and post-translational level. It would be informative to determine the contribution of the impact of Vpr on SPI1 expression on the reduced PU.1 protein levels. In that context, the impact of Vpr on endogenous PU.1 expression could be assessed +/- MG132.

5- The authors claim that Vpr-mediated downmodulation of PU-1 regulated gene expression is mediated by direct Vpr-PU.1 interaction, resulting in accelerated proteasomal degradation of PU.1. They propose a model in which interactions amongst PU.1, Vpr and DCAF1 promote the ubiquitination of PU.1 via the associated CUL4A ubiquitin ligase complex with resultant proteasomal degradation. These conclusions are based on the data obtained with Vpr Q65R and DCAF1-depleted cells (Figure 7A). The Q65R substitution resulted in the loss of DCAF1 association and disrupted the formation of the Vpr- PU.1 complex (Figure 7A Lane 6). However, this mutation had not impact on Vpr-mediated PU.1 degradation (Lane 18 vs lane 14), how the authors explain this seemingly contradictory result? If Q65R is unable to recruit the E3 ligase and doesn't bind PU.1, then how can the authors explain the observed PU.1 degradation? (a quantification of PU.1 pull-down/input, would be very helpful). Similarly, DCAF1 depletion reduced Vpr-PU.1 interaction (Lane 10), but had no impact on Vpr-mediated PU.1 degradation (Lane 22 vs lane 20). Again, how this result can be reconciled with the authors working model? Are these phenotypes linked to overexpression? Experiments aimed at measuring the impact of the Q65R mutation and DCAF1 depletion on PU.1 levels (in the context of endogenous PU.1 expression

in MDMs) would strengthen the manuscript.

Minor comments:

6- Statistical significance should be shown in Figure 2E.

7- In Figure 3C, the statistical significance for IFI6 is missing.

8- In Figure 4G as well as 5C and E, based on the figure legend, the % of PU.1+ cells per transfected cells (FLAG+) must be shown.

Reviewer #4 (Remarks to the Author):

Vpr is necessary for HIV-1 to replicate *in vivo*, closely related to disease progression in patients. The molecular mechanisms underlying the function of Vpr in HIV-1 infectivity are urgent to be elucidated. In this manuscript, Virgilio et al. reveal that Vpr counters an innate immune response to HIV-infection in macrophages via interactions among PU.1, Vpr, and DCAF1 to promote the ubiquitylation of PU.1. This mechanism provides a new restriction factor in macrophages against Vpr for HIV-1 treatment. However, several issues need to be addressed.

Major comments:

1. As shown in Figure 1D and 1E, there is no significant differences among experimental groups (Uninfected, 89.6 Vpr-null, and 89.6 WT-Vpr) which implies that the transcriptomic profiles are similar. It is in contradiction with the differential expression result in Figure 1F.

2. What are the differences of the clusters identified in Figure 1E? Whether they exhibit different expression profiles related to virion assembly and spread?

3. The expression of genes (ISG15, LY96, IFI6 and IFITM3) involved in the TLR and IFN-I signaling did not significantly decrease in infected MDMs treated with Δ vpr 89.6 HIV than wt 89.6 HIV (Figure 3C). LY96 and IFITM3 even showed higher expression in infected MDMs treated with Δ vpr 89.6 HIV than wt 89.6 HIV. Moreover, ISG15, LY96 and IFI6 seemed to have higher expression in uninfected MDMs treated with wt 89.6 HIV than gag/tat+. It is cursory to draw a conclusion that "Vpr suppresses PU.1 regulated genes implicated in Toll-like receptor and IFN-I responses" based on these results.

4. The working model of PU.1 and TET2 interacting with Vpr and DCAF1 in macrophages should be validated in MMDs.

Minor comments:

5. The circle for GO terms in Figure 2D should be highlighted.

6. The statistic should be provided in Figure 1E.

7. Line 237: Figure 4I should be cited instead of Supplemental Figure 3A.

8. Line 253-255: "Supplemental Figure 3B" and "Supplemental Figure 3C" should be cited.

9. Line 298: Reference should be cited.

10. Line 304: The full name of the "IP" (immunoprecipitation) should be provided.

11. Line 374: wrong citation for the figure.

12. Legend of figure 2D: (D) Biological processes associated with the PU.1 targeted genes from (C)?

We thank the reviewers for their insightful comments and suggestions for new experiments. The revised manuscript is significantly improved in the strength of the findings as well as the communication of the results. We have fully responded to each of the reviewers' comments in an itemized fashion below:

Reviewer 1:

Reviewer Summary: "Vpr is an enigmatic protein required for efficient spread of HIV from macrophages to T cells". VPR's pleiotropic effects have been hard to nail down because it is widely accepted that they are mediated via a pleiotropic mechanism of ubiquitination and proteasomal degradation targeting various selected proteins. This study presents and examines the significant and noteworthy finding that PU.1 is a target of VPR mediated degradation. PU.1 is a myeloid specific transcription factor that is required for efficient induction of the host innate immune response to HIV. By degradation of PU.1 VPR can effectively attenuate transcription of many innate response transcripts and mute the host's defensive response to HIV infection. scRNA seq and bioinformatic analyses suggest the hypothesis that PU.1 is a key target of VPR degradation. This work tests that notion using a number of HIV and other lentiviral analog constructs with contrasting WT and inactive versions of VPR (and analogs). Flow cytometry, Western Blot and pull down experiments quantify protein levels and interactions. The work meets the new standard for the field in which omics observations are "ground truthed". Experimentally, it is well designed and executed, yielding convincingly interpretable data. The methodology is sound and the information detail adequate for repetition. As always, good work leaves the reader wanting more:

Comment 1: While the whole premise of the proposed mechanism is that the VPR-mediated proteasome degradation of PU.1 and TET2 results in the down regulation of macrophage innate response gene transcripts, there is very little investigation/quantification of the ubiquitination of these targets or the specific proteins involved in their degradation. While convincing evidence of the VPR dependency of reduced levels of these key proteins is presented (closely linked to transcriptomic quantification of responsive genes), the work stops there.

Comment 1 Response:

In the revised manuscript, we provide extensive evidence for the novel finding that: (1) Vpr expression decreases the expression of PU.1 dependent genes (Figures 2 and 3); (2) Vpr physically interacts with PU.1 (Figure 6), and (3) stimulates the proteasomal degradation of PU.1 (Figure 7). We furthermore show that complex formation and degradation depend on DCAF1 (Figure 6 and 7). Finally, we show that Vpr-dependent degradation of PU.1 is an evolutionarily conserved function of Vpr (Figures 4 and 5).

In the revised manuscript, we validated the transcriptomic data showing that Vpr suppresses PU.1 dependent gene expression. This was done by comparing protein expression levels in "wild type" HIV infected versus Vpr-null HIV infected primary macrophages. Because not all the cells in the culture were infected, confocal microscopy was used to identify infected cells and quantify expression levels of PU.1 dependent gene products (Figure 3D-I).

We also extended results from cell line models to primary macrophages to confirm that Vpr forms a complex with PU.1 and promotes degradation of PU.1 in primary HIV-exposed macrophages (Figures 6 and 7). Moreover, the degradation we observed was most likely mediated by ubiquitylation and proteasomal degradation because it was reversible by proteasome inhibitors (Figure 7). Degradation also required interaction with DCAF1 based results showing that a Vpr mutant defective an interacting with DCAF1 was defective at

degradation (Figure 7). All of these results point to a model in which Vpr binds PU.1 to connect it to the DCAF1-Cul4 E3 ubiquitin ligase complex, promoting PU.1 ubiquitylation and proteasomal degradation. Despite this being the most likely model, we were unable to detect ubiquitylated intermediates of PU.1 using antibodies directed against ubiquitin in western blot analysis of PU.1 immunoprecipitates from Vpr expressing cells. We also obtained negative results when transfecting HA-tagged ubiquitin and using antibodies directed against PU.1 in western blot analysis of ubiquitin immunoprecipitates. These negative results may be due to the labile nature of such intermediates. We discuss this limitation to our study in the revised manuscript (Lines 467, 521-526).

Prior studies have demonstrated that Vpr counteracts restriction factors that target Env for degradation. In the revised manuscript we provide a new experiment allowing us to come full circle by showing that silencing PU.1 in primary HIV-infected macrophages increased HIV Env expression compared with an HIV that lacked Vpr (Figure 8). This is consistent with our model that Vpr counteracts restriction factors by inhibiting PU.1-dependent transcription of two restriction factors targeting HIV Env (macrophage mannose receptor and IFITM3).

Reviewer 2:

Reviewer summary: The manuscript by Virgilio et al studies the role of HIV-1 Vpr in monocytes by examining the impact on cellular transcription both in infected cells and in bystander cells within the infected pool. They center their analysis on the downregulation of transcription factor PU.1 which alters upregulation of innate immune genes ISG15, LY96 and IFI6. The authors suggest a model whereby Vpr inhibits the induction of innate antiviral immune programs by binding to and inducing DCAF-1 dependent degradation of PU.1. The hypothesis presents a novel mechanism and explains a dominant transcriptional changes that occur only in bystander cells, not in the infected cells. Overexpression and knockdown experiments to further test their hypothesis are well performed, and evidence for a direct interaction with the PU.1 transcription factor is convincing with co-IP studies. The study supports a novel mechanism of action of Vpr that can prevent interferon responses in infected cells, by directly downregulating a key required transcription factor, and shows that this is a conserved activity of many alleles of vpr from SIV, HIV-2.

Comment 1: There appears to be a modest benefit to the WT Vpr expressing infected cells which have higher HIV gene expression. Nonetheless, the overall phenotype regarding how this impacts viral replication or persistence is still unclear. The discussion may benefit from some additional attention to how the authors believe this activity benefits the virus as the role of Vpr remains somewhat enigmatic.

Comment 1 Response:

As discussed in our response to Reviewer 1, the revised manuscript contains new data that further illuminates how Vpr-dependent degradation of PU.1 impacts viral spread in macrophages. In prior publications, we and others reported that Vpr expression over comes restriction factor(s) that limit HIV spread in primary macrophages by reducing HIV Env expression levels. Remarkably, both restriction factors that target Env (mannose receptor and IFITM3) depend on PU.1 for expression. Thus, the discovery that Vpr degrades PU.1 (reported here) explains the observed reduction in expression of both of these factors. To fully close the circle, we asked whether reduction of PU.1 rescued Env expression in the absence of Vpr. This was examined through the design of an HIV vector that allowed silencing of PU.1 in Vpr-null HIV-infected macrophages. Indeed, we found that silencing PU.1 expression led to increased HIV Env expression compared to macrophages infected with a Vpr-null HIV that lacked the PU.1 shRNA (Figure 8). These data help support our model that Vpr acts through its effects on

PU.1 to counteract factors that restrict HIV Env and thereby limit spread of HIV. The revised manuscript was modified to include a discussion of these new findings (Lines 532-534).

Minor note:

Figure 4B. In the bystander GFP- cells there appears to be some downregulation of PU.1 as well when compared to the vpr null. Is this evidence of downregulation of PU.1 in cells that may not yet be expressing HIV, perhaps through virus particle associated vpr? Analysis of the flow plots with 2 colors GFP vs PU.1 may be especially informative here.

Response:

That is an interesting point. However, a deeper analysis revealed that MDM treated with replication defective virus did not have significant downmodulation of PU.1 in the bystander cells after five days in culture (New Supplementary Figure 3A). This is not surprising given expected turnover of virion-associated Vpr after five days in culture. In a new experiment, we exposed primary macrophages to Vpr-containing virions for five hours and demonstrate that this treatment was sufficient to degrade PU.1 (Figure 7). Degradation was reversible by proteasome inhibitors and depended on interactions between Vpr and DCAF-1 [VPR mutant Q65R, which is defective at interacting with DCAF1 did not degrade PU.1 in MDM treated with viral particles for five hours (Figure 7)]. The short five-hour exposure is not expected to result in productive infection and Vpr protein expression, which would take a minimum of about 24 hours and 2 or 3 days to complete¹. Thus, we were able to validate the reviewer's hypothesis that virus particle associated Vpr can mediate PU.1 degradation in bystander cells. This was discussed in the revised manuscript (Lines 424-440, 473-475).

Reviewer 3:

Reviewer summary: In this study, Virgilio et al., used single cell RNAseq to capture the transcriptional changes mediated by Vpr during HIV-1-infection of macrophages. The authors previously demonstrated that Vpr counteracts accelerated degradation of the Env glycoprotein initiated by the mannose receptor (MR) by suppressing expression of the MR gene, MRC1. However, the mechanism by which Vpr suppresses MRC1 expression wasn't known. Here they show that Vpr reduces the levels of PU.1, a transcription factor regulating many essential macrophage genes, including MRC1. These timely results suggest a model in which Vpr could reprogram HIV-1-infected macrophages by targeting PU.1. Accordingly, Vpr was shown to disrupt the expression of several PU.1 associated antiviral factors, including MRC1, but also IFITM3, ISG15, IFI6 and LY96. Importantly, the capacity of Vpr to reduce PU.1 levels was conserved among HIV-1, HIV-2 and SIV.

Comment 1: While data presented suggest that PU.1 and DCAF1 form a complex with Vpr, it remains unclear whether PU.1 degradation requires direct Vpr-PU.1 interaction and the recruitment of the DCAF1-Cul4 E3 ubiquitin ligase complex. Moreover, direct experiments connecting Vpr-mediated PU.1 degradation to its capacity to regulate gene expression and favor HIV replication in MDMs are needed.

Comment 1 Response:

To address the reviewer's concern, we performed additional experiments that directly connect Vpr-mediated PU.1 degradation with enhanced HIV replication in MDMs. This was accomplished through the design of an HIV vector that allowed silencing of PU.1 in Vpr-null HIV-infected macrophages. Using this vector, we found that silencing PU.1 led to increased HIV Env expression compared to macrophages infected with a Vpr-null HIV that contained a scrambled control shRNA (Figure 8). These data help support the model that Vpr acts through its effects on

PU.1 to counteracts factors that target HIV Env and thereby limit spread of HIV. The revised manuscript was modified to include a discussion of these new findings (Lines 532-534).

Other new experiments provided in the revised manuscript further strengthen these conclusions; for example, we extended results initially performed in cell line models to primary HIV infected macrophages. Specifically, we provide validation that Vpr forms a complex with PU.1 and promotes proteasomal degradation of PU.1 in primary HIV-infected macrophages (Figure 6 and 7). Additionally, see response to comment 2 below for other reviewer-requested studies validating our conclusions in primary macrophages.

Comment 2: In Figure 2E and 3C, the authors show that Vpr reduces the expression of several transcripts involved in TLR and IFN-I signaling, such as ISG15, IFITM3 and IFI6 that contribute to host innate immune response to HIV. How this impacts the level of the proteins encoded by these transcripts was not reported. Evaluation of the levels of the proteins encoded by these transcripts (in HIV-1 infected and uninfected bystander MDMs) would confirm the impact of Vpr on host innate immune response.

Comment 2 Response:

To address this concern, we performed quantitative confocal microscopy and determined that infected (multi-nucleated Gag-positive primary macrophages) had lower levels of PU.1-dependent innate immune response gene products than macrophages infected with a Vpr-null HIV (Figure 3D-I). Furthermore, we quantified the protein levels in uninfected bystander (Gag-negative) MDMs and found reduced IFITM3 and ISG15 associated with Vpr exposure. This result confirms our scRNA-seq findings suggesting a bystander effect of Vpr on uninfected, virally exposed cells (Figure 3F, I). New data showing that relatively short (five hour) exposure to virion-associated Vpr causes detectable levels of PU.1 degradation in macrophages (Figure 7A & B, D & E) helps explain this result.

Comment 3: The author's main conclusions are that Vpr reprograms HIV-1-infected MDMs gene expression by targeting PU.1 and that contributes to disrupt expression of several antiviral factors (MCR1, ISG15, IFITM3, LY96 and IFI6) restricting HIV infection and spread. Evaluating the impact of PU.1 depletion on the capacity of Vpr to modulate antiviral gene expression in MDMs therefore favoring HIV replication would strengthen the conclusions of the manuscript.

Comment 3 Response:

As discussed above in our response to comment 1, prior studies have shown that Vpr is needed to counteract restriction of HIV Env expression and promote viral spread. In the revised manuscript we came full circle by showing that silencing PU.1 in HIV-Vpr-null-infected macrophages rescued HIV Env expression (Figure 8). This experiment was technically difficult because we and others have found that siRNAs and shRNAs can be sensed by macrophages leading to upregulation of antiviral factors that limit viral infection and spread. To avoid the si/shRNA response, which has limited our ability to infect siRNA-treated macrophages with HIV, we constructed an HIV that harbored shRNA cassettes (control scrambled shRNA and shRNAs directed against PU.1) in the Vpr open reading frame. This strategy generated sufficient infected cell numbers to conclude that silencing PU.1 rescued Env expression compared to control, which supports our model. However, evaluation of silenced primary macrophages to assess expression levels of antiviral factors in these cells was not feasible because all HIVs harboring shRNA cassettes, including the negative control, scrambled shRNA, did not spread well enough to acquire the cell numbers needed for this more extensive analysis.

Although we were unable to measure the impact of silencing PU.1 on interferon-induced genes (e.g. *ISG15*, *IFITM3*, *IFI6*) which are all upregulated in response to shRNAs, we instead

examined whether silencing PU.1 alone could reduce mannose receptor, which is naturally highly expressed in macrophages and does not rely on innate immune sensing for upregulation. In this case, we were able to successfully demonstrate a reduction of mannose receptor with PU.1 silencing (Supplementary Figure 2B).

Comment 4: The authors previously demonstrated that Vpr counteracts accelerated degradation of Env by reducing MCR1 expression. It is unclear why the authors didn't evaluate the impact of PU.1 depletion on Vpr's ability to reduce Env expression. Surprisingly, the only experiment evaluating the impact of Vpr on PU.1 levels in MDMs was performed using an Env-defective virus (Figure 4A,B,C). This looks like a straightforward experiment for this group and it would greatly benefit the manuscript.

Comment 4 Response:

As discussed above, this experiment was successfully performed in the revised manuscript (Figure 8).

Comment 5: SPI1 mRNA levels, encoding for PU.1 were also lower in the presence of Vpr (Figure 2B and 3B), suggesting that PU.1 downmodulation could occur at the transcriptional and post-translational level. It would be informative to determine the contribution of the impact of Vpr on SPI1 expression on the reduced PU.1 protein levels. In that context, the impact of Vpr on endogenous PU.1 expression could be assessed +/- MG132.

Comment 5 Response:

We agree with the reviewer that Vpr likely acts at both the transcriptional and post-translational levels. To address the reviewer's concern, we performed a new set of experiments that clearly demonstrated degradation of endogenous PU.1 in primary macrophages following treatment of wild type HIV in a manner that was reversible by proteasome inhibitor treatment (MG132) as suggested (Figure 7 D, E). Thus, at least after short incubations (five hours), MG132 seemed to near-fully restore PU.1 levels. We do expect that after longer time points in primary cells that transcriptional mechanisms would lead to additional downmodulation of PU.1 in fully infected cells. Nevertheless, this experiment confirms the existing data in the manuscript performed in HEK 293T cells exogenously expressing both PU.1 and Vpr (Figure 7C).

Comment 6: The authors claim that Vpr-mediated downmodulation of PU.1 regulated gene expression is mediated by direct Vpr-PU.1 interaction, resulting in accelerated proteasomal degradation of PU.1. They propose a model in which interactions amongst PU.1, Vpr and DCAF1 promote the ubiquitination of PU.1 via the associated CUL4A ubiquitin ligase complex with resultant proteasomal degradation. These conclusions are based on the data obtained with Vpr Q65R and DCAF1-depleted cells (Figure 7A). The Q65R substitution resulted in the loss of DCAF1 association and disrupted the formation of the Vpr- PU.1 complex (Figure 7A Lane 6). However, this mutation had no impact on Vpr-mediated PU.1 degradation (Lane 18 vs lane 14), how the authors explain this seemingly contradictory result? If Q65R is unable to recruit the E3 ligase and doesn't bind PU.1, then how can the authors explain the observed PU.1 degradation? (a quantification of PU.1 pull-down/input, would be very helpful). Similarly, DCAF1 depletion reduced Vpr-PU.1 interaction (Lane 10), but had no impact on Vpr-mediated PU.1 degradation (Lane 22 vs lane 20). Again, how this result can be reconciled with the authors working model? Are these phenotypes linked to overexpression? Experiments aimed at measuring the impact of the Q65R mutation and DCAF1 depletion on PU.1 levels (in the context of endogenous PU.1 expression in MDMs) would strengthen the manuscript.

Comment 6 Response:

The reviewer's concern that overexpression of PU.1 might be limiting our ability to accurately detect degradation is very reasonable. In addition, the experimental protocol was optimized for immunoprecipitation rather than for assessing protein degradation. To assess degradation accurately, we found that it is important to utilize homogeneous cell cultures or single cell

approaches (e.g. Figures 4 and 5) that can ensure expression of both Vpr and PU.1 in the sampled population. In addition to overexpressing PU.1, the cells used for immunoprecipitation contained heterogeneous mixtures of transfected and untransfected cells - including some transfected cells that contained PU.1 but not Vpr constructs. Thus, it is not surprising that western blot analysis of bulk cell lysates was not sufficiently sensitive to detect degradation in all experiments.

To address the reviewer's concern, we performed new experiments in primary macrophages as requested. The experiments were optimized to ensure that a majority of the cells in the culture were exposed to virion associated HIV Vpr. Under these conditions, we confirmed that Vpr promotes degradation of PU.1 through a mechanism that was inhibitable by the proteasome inhibitor MG132 (Figure 7 D, E). We also confirmed that degradation depended on an interaction between Vpr and DCAF1. This conclusion was based on the observation that the Vpr Q65R mutant that is defective at interacting with DCAF1 was also defective at promoting PU.1 degradation (Figure 7 A, B).

Minor comments:

7- *Statistical significance should be shown in Figure 2E.*

- Statistical significance was added.

8- *In Figure 3C, the statistical significance for IFI6 is missing. – not missing, just 0!*

- The p-value is small enough that it was rounded to 0 using the Wilcoxin Rank Sum function in our single-cell analysis program, LIGER. We have instead adjusted the p-value to be less than the smallest measurable p-value.

9- *In Figure 4G as well as 5C and E, based on the figure legend, the % of PU.1+ cells per transfected cells (FLAG+) must be shown.*

- Y-axes have been re-labeled to reflect this.

Reviewer 4:

Reviewer summary: Vpr is necessary for HIV-1 to replicate in vivo, closely related to disease progression in patients. The molecular mechanisms underlying the function of Vpr in HIV-1 infectivity are urgent to be elucidated. In this manuscript, Virgilio et al. reveal that Vpr counters an innate immune response to HIV-infection in macrophages via interactions among PU.1, Vpr, and DCAF1 to promote the ubiquitylation of PU.1. This mechanism provides a new restriction factor in macrophages against Vpr for HIV-1 treatment.

Major comments:

Comment 1: As shown in Figure 1D and 1E, there is no significant differences among experimental groups (Uninfected, 89.6 Vprnull, and 89.6 WT-Vpr) which implies that the transcriptomic profiles are similar. It is in contradiction with the differential expression result in Figure 1F.

Comment 1 Response:

We clarified in the revised manuscript that we used LIGER to deliberately align datasets despite significant transcriptomic differences between donors and sample types. LIGER allows us to first find a set of shared cell types, then look for differences within the groups, allowing us to integrate all the sample types while also allowing for the detection of transcriptional differences between virus treatment types (Lines 111-118).

Comment 2: What are the differences of the clusters identified in Figure 1E? Whether they exhibit different expression profiles related to virion assembly and spread?

Comment 2 Response:

The cluster descriptions are included in the figure legend. Based on gene expression patterns in each of the clusters, we determined the main cluster to be pro-inflammatory macrophages (Cluster 0) as well as a minor population of cycling cells (Cluster 1). The remaining clusters have been removed from downstream analysis because they were found to contain low gene counts. The two remaining clusters do not appear to have an infection phenotype because the infected cells are distributed across both the clusters. This was clarified in the revised manuscript (Lines 114-118).

Comment 3: The expression of genes (ISG15, LY96, IFI6 and IFITM3) involved in the TLR and IFN-I signaling did not significantly decrease in infected MDMs treated with Δ vpr 89.6 HIV than wt 89.6 HIV (Figure 3C). LY96 and IFITM3 even showed higher expression in infected MDMs treated with Δ vpr 89.6 HIV than wt 89.6 HIV. Moreover, ISG15, LY96 and IFI6 seemed to have higher expression in uninfected MDMs treated with wt 89.6 HIV than gag/tat+. It is cursory to draw a conclusion that “Vpr suppresses PU.1 regulated genes implicated in Toll-like receptor and IFN-I responses” based on these results.

Comment 3 Response:

We agree and that is indeed the point that we are trying to make. We observed higher expression of innate immune response genes in the absence of Vpr. Our data supports the model that Vpr counters an upregulation that would otherwise occur by targeting PU.1. In the revised manuscript, we strengthened these findings by performing confocal microscopy on infected cells and showing that multinucleated Gag positive Vpr expressing primary macrophages express lower levels of innate immune response proteins (Figure 3D-I).

Comment 4: The working model of PU.1 and TET2 interacting with Vpr and DCAF1 in macrophages should be validated in MDMs.

Comment 4 Response:

As indicated above, to address this concern and strengthen the manuscript, we verified that Vpr and DCAF1 form a complex with PU.1 in primary HIV infected macrophages (Figure 6C). Unfortunately, despite performing immunoprecipitations from 60 million MDMs, we were unable to sufficiently detect TET2 in the in the IP. While others have reported the interaction between PU.1 and TET2² that confirms our results in HEK 293T cells, the inability to detect this interaction in MDMs is an important limitation that was discussed in the revised manuscript. The limitation could have resulted from lower expression of TET2 in MDM than in 293T cell lines or because of stronger interaction with PU.1 than Vpr. (PU.1 was FLAG-tagged in 293T cells to facilitate immunoprecipitation but was not tagged in primary macrophage experiments.)

Moreover, in the revised manuscript, we strengthened our conclusions by showing; (1) degradation of PU.1 in HIV-infected primary macrophages is blocked by proteasome inhibitors (Figure 7 D, E), (2) degradation of PU.1 in HIV-infected primary macrophages required an intact DCAF1 interaction domain in Vpr (Figure 7 A, B), and (3) silencing PU.1 in HIV-infected primary macrophages rescued Env expression (Figure 8C).

Minor comments:

Comment 5. The circle for GO terms in Figure 2D should be highlighted.

- We have highlighted the GO terms in the figure.

Comment 6. The statistic should be provided in Figure 1E. – not sure what statistics...maybe number of infected cells in each population?

- Supplemental Table 1 has been added to include the total number of cells in each cluster or by infection type before and after segregating cells based on gag/tat-positivity. Some of these numbers have also been added to the text. The percent of gag⁺ cells for each infected sample by donor is provided in Supplemental Table 2 (previously Supplemental Table 1) (Lines 122-123).

Comment 7. Line 237: Figure 4I should be cited instead of Supplemental Figure 3A.

- Reference to Supplemental Figure 3A was to show the HIV construct used in Figure 4I. We apologize for the confusion. We agree the results referenced in the text are shown in Figure 4I. Reference to Supplemental Figure 3A was to point the audience to the HIV construct used in generating Figure 4I, which was not shown in the main figure or used to generate the remaining data in Figure 4 but was used to generate the remaining data in Supplemental Figure 3. We used a different HIV construct not shown in the main figure because it has a fluorescent protein that allows us to measure HIV expression and we have a vpr-wt and vpr-null version of the construct that was relevant to Figure 4I.

Comment 8. Line 253-255: “Supplemental Figure 3B” and “Supplemental Figure 3C” should be cited.

- This has been corrected.

Comment 9. Line 298: Reference should be cited.

- References have been added.

Comment 10. Line 304: The full name of the “IP” (immunoprecipitation) should be provided.

- This has been corrected.

Comment 11. Line 374: wrong citation for the figure.

- This has been corrected.

Comment 12. Legend of figure 2D: (D) Biological processes associated with the PU.1 targeted genes from (C)?

- This has been corrected.

References

1. Mashiba, M., Collins, D. R., Terry, V. H. & Collins, K. L. Vpr overcomes macrophage-specific restriction of HIV-1 Env expression and virion production. *Cell Host Microbe* **16**, 722–735 (2014).
2. de la Rica, L. *et al.* PU.1 target genes undergo Tet2-coupled demethylation and DNMT3b-mediated methylation in monocyte-to-osteoclast differentiation. *Genome Biol.* **14**, 1–21 (2013).

REVIEWER COMMENTS

Reviewer #1 (Remarks to the Author):

Reviewer Summary: The current manuscript "HIV-1 Vpr combats the PU.1-driven antiviral response in primary human macrophages," Virgilio and colleagues is well written, clear to follow and offers convincing evidence of the role of PU.1 in the activity of Vpr to down regulate innate defenses of macrophages infected with HIV-1.

This reviewer found the responses to the original critique to be adequate and recommend publication.

Comment #1.

One lingering issue that is discussed in the manuscript and to some length in the previous reviews is the effect of HIV-1 on bystander cells, and the suppression of innate defenses in bystander cells (defined as cells not making gag protein). Lim et al. (PMID: 35166645) published, using single cell analysis of HIV-1 infected THP-1 cells, that bystander cells respond to HIV by modifying their transcriptome. Further, they showed that macrophages containing pre-integration HIV-1 cDNA complexes exhibit different transcriptome profiles than control cells (not exposed to virus) or cells containing provirus. The question is: Would the possible presence of pre-integration HIV-1 cDNA complex containing cells, which would be included in the bystander group as defined by absence of gag, possibly complicate the interpretation of Vpr effects in bystander cells? I personally don't think so, but I would appreciate the authors considering this possibility. In any case the Lim paper should be referenced since it first described HIV-1 effects on bystander cells using scRNA seq.

Reviewer #2 (Remarks to the Author):

The revised manuscript by Virgilio is improved and provides additional evidence that PU.1 downregulation through a non-Vpr mediated method, can also enhance HIV Env expression. They also demonstrate that the Vpr in virus particles is sufficient to downregulate PU.1 in a proteasome-dependent manner. Both points strengthen an already compelling manuscript, shedding additional light on how Vpr enhances growth in monocyte macrophages.

Reviewer #3 (Remarks to the Author):

see attached

Reviewer #4 (Remarks to the Author):

The authors have addressed the majority of the comments and provided detailed responses. Just a minor comment

- In Figure 1E, the authors should provide some marker genes/features for annotating the identified clusters of MDMs (Cluster 0: pro-inflammatory macrophages and Cluster 1: cycling cell).

Reviewer #3 (Attachment):

Reviewer summary: In this study, Virgilio et al., used single cell RNAseq to capture the transcriptional changes mediated by Vpr during HIV-1-infection of macrophages. The authors previously demonstrated that Vpr counteracts accelerated degradation of the Env glycoprotein initiated by the mannose receptor (MR) by suppressing expression of the MR gene, MRC1. However, the mechanism by which Vpr suppresses MRC1 expression wasn't known. Here they show that Vpr reduces the levels of PU.1, a transcription factor regulating many essential macrophage genes, including MRC1. These timely results suggest a model in which Vpr could reprogram HIV-1-infected macrophages by targeting PU.1. Accordingly, Vpr was shown to disrupt the expression of several PU.1 associated antiviral factors, including MRC1, but also IFITM3, ISG15, IFI6 and LY96. Importantly, the capacity of Vpr to reduce PU.1 levels was conserved among HIV-1, HIV-2 and SIV.

Reviewer Comment 1: While data presented suggest that PU.1 and DCAF1 form a complex with Vpr, it remains unclear whether PU.1 degradation requires direct Vpr-PU.1 interaction and the recruitment of the DCAF1-Cul4 E3 ubiquitin ligase complex. Moreover, direct experiments connecting Vpr-mediated PU.1 degradation to its capacity to regulate gene expression and favor HIV replication in MDMs are needed.

Authors response:

To address the reviewer's concern, we performed additional experiments that directly connect Vpr-mediated PU.1 degradation with enhanced HIV replication in MDMs. This was accomplished through the design of an HIV vector that allowed silencing of PU.1 in Vpr null HIV-infected macrophages. Using this vector, we found that silencing PU.1 led to increased HIV Env expression compared to macrophages infected with a Vpr-null HIV that contained a scrambled control shRNA (Figure 8). These data help support the model that Vpr acts through its effects on PU.1 to counteract factors that target HIV Env and thereby limit spread of HIV. The revised manuscript was modified to include a discussion of these new findings (Lines 532-534).

Other new experiments provided in the revised manuscript further strengthen these conclusions; for example, we extended results initially performed in cell line models to primary HIV infected macrophages. Specifically, we provide validation that Vpr forms a complex with PU.1 and promotes proteasomal degradation of PU.1 in primary HIV-infected macrophages (Figure 6 and 7). Additionally, see response to comment 2 below for other reviewer-requested studies validating our conclusions in primary macrophages.

Reviewer response: The new data in MDMs confirming the capacity of Vpr to promote PU.1 degradation are convincing and certainly contribute to strengthen the current manuscript. However, direct experiments connecting Vpr-mediated PU.1 degradation to its capacity to regulate gene expression and favor HIV replication in MDMs are still missing (see comment 2, 3 and 4 below).

Reviewer Comment 2: In Figure 2E and 3C, the authors show that Vpr reduces the expression of several transcripts involved in TLR and IFN-I signaling, such as ISG15, IFITM3 and IFI6 that contribute to host innate immune response to HIV. How this impacts the level of the proteins encoded by these transcripts was not reported. Evaluation of the levels of the proteins encoded by these transcripts (in HIV-1 infected and uninfected bystander MDMs) would confirm the impact of Vpr on host innate immune response.

Authors response:

To address this concern, we performed quantitative confocal microscopy and determined that infected (multi-nucleated Gag-positive primary macrophages) had lower levels of PU.1-dependent innate immune response gene products than macrophages infected with a Vpr-null HIV (Figure 3D-I). Furthermore, we quantified the protein levels in uninfected bystander (Gag-negative) MDMs and found reduced IFITM3 and ISG15 associated with Vpr exposure. This result confirms our scRNA-seq findings suggesting a bystander effect of Vpr on uninfected, virally exposed cells (Figure 3F, I). New data showing that relatively short (five hour) exposure to virion-associated Vpr causes detectable levels of PU.1 degradation in macrophages (Figure 7A & B, D & E) helps explain this result.

Reviewer response: Confocal microscopy does not represent a robust and sensitive technique to measure protein levels. Moreover, the quantification was done on only few cells (between 17 to 27 cells total). Flow cytometry and Western blotting represent more quantitative techniques to measure IFITM3, ISG15 and IFI6 protein levels and these techniques are already used by the authors in the current manuscript to measure PU.1 and MR protein levels in infected MDMs. The system used to measure the impact of Vpr on PU.1 protein levels in Figure 4 by flow cytometry, could therefore be also used to measure IFITM3, ISG15 and IFI6 protein levels. Alternatively, their protein levels could be assessed in infected MDM by Western blot using the condition used in Figure 7. Since Vpr-mediated PU.1 degradation is observed under these conditions, this should allow the authors to measure the impact of Vpr in IFITM3, ISG15 and IFI6 protein levels. Notably, the same group previously measured IFITM3 expression in infected MDMs by Western blot (Lubow *et al.*, 2020, eLife).

Reviewer Comment 3: The author's main conclusions are that Vpr reprograms HIV-1-infected MDMs gene expression by targeting PU.1 and that contributes to disrupt expression of several antiviral factors (MCR1, ISG15, FITM3, LY96 and IFI6) restricting HIV infection and spread. Evaluating the impact of PU.1 depletion on the capacity of Vpr to modulate antiviral gene expression in MDMs therefore favoring HIV replication would strengthen the conclusions of the manuscript.

Authors response:

As discussed above in our response to comment 1, prior studies have shown that Vpr is needed to counteract restriction of HIV Env expression and promote viral spread. In the revised manuscript we came full circle by showing that silencing PU.1 in HIV-Vpr-null-infected macrophages rescued HIV Env expression (Figure 8). This experiment was technically difficult because we and others have found that siRNAs and shRNAs can be sensed by macrophages leading to upregulation of antiviral factors that limit viral infection and spread. To avoid the si/shRNA response, which has limited our ability to infect siRNA-treated macrophages with HIV, we constructed an HIV that harbored shRNA cassettes (control scrambled shRNA and shRNAs directed against PU.1) in the Vpr open reading frame. This strategy generated sufficient infected cell numbers to conclude that silencing PU.1 rescued Env expression compared to control, which supports our model. However, evaluation of silenced primary macrophages to assess expression levels of antiviral factors in these cells was not feasible because all HIVs harboring shRNA cassettes, including the negative control, scrambled shRNA, did not spread well enough to acquire the cell numbers needed for this more extensive analysis.

Although we were unable to measure the impact of silencing PU.1 on interferon-induced genes (e.g. ISG15, IFITM3, IFI6) which are all upregulated in response to shRNAs, we instead examined whether silencing PU.1 alone could reduce mannose receptor, which is naturally highly expressed in macrophages and does not rely on innate immune sensing for upregulation. In this case, we were able to successfully demonstrate a reduction of mannose receptor with PU.1 silencing (Supplementary Figure 2B).

Reviewer response: The data presented in Figure 8 are confirming the impact of PU.1 on HIV-1 Env and MR expression in MDMs. However, they do not directly address the impact of Vpr-mediated PU.1 degradation on its capacity to modulate antiviral gene expression and favor HIV replication, as these experiments were only done with a Vpr null virus. Previous studies by the same group (Mashiba *et al.*, 2014, Cell host Microbe; Collins *et al.*, 2015 Plos Pathog; Lubow *et al.*, 2020,eLife) and other groups (Klockow *et al.*, 2013, Virology; Zhao *et al.*, 2015 Nat Com; Kyei *et al.*, 2015, Cell Host Microbe; Krapp *et al.*, 2016, Cell Host Microbe) used si/shRNA to silence gene expression in MDMs prior HIV-1-infection. Notably, using shRNA-expressing lentiviral construct optimized to limit antiviral responses in MDMs, the same group previously silenced DCAF-1 and MR in MDMs before successfully infecting them with WT or Vpr- virus (Mashiba *et al.*, 2014, Cell host Microbe; Collins *et al.*, 2015 Plos Pathog; Lubow *et al.*, 2020,eLife) . Using this system, they determined the impact of DCAF-1 or MR silencing on the capacity of Vpr to enhance Env expression, virus production and replication. Similar experiments could be done here to directly evaluate the impact of PU.1 silencing on the capacity of Vpr to modulate antiviral gene expression and favor HIV-1 replication. To directly assess the impact of PU.1 degradation in the capacity of Vpr to favor HIV replication, the authors should measure viral spread (in MDMs or in coculture with T cells) or at least virion production/infectivity as previously published (Mashiba *et al.*, 2014, Cell host Microbe; Collins *et al.*, 2015 Plos Pathog; Lubow *et al.*, 2020, eLife), not only Env expression.

Reviewer Comment 4: The authors previously demonstrated that Vpr counteracts accelerated degradation of Env by reducing MCR1 expression. It is unclear why the authors didn't evaluate the impact of PU.1 depletion on Vpr's ability to reduce Env expression. Surprisingly, the only experiment evaluating the impact of Vpr on PU.1 levels in MDMs was performed using an Env-defective virus (Figure 4A,B,C). This looks like a straightforward experiment for this group and it would greatly benefit the manuscript.

Authors response:

As discussed above, this experiment was successfully performed in the revised manuscript (Figure 8).

Reviewer response: This experiment indeed confirms the impact of PU.1 depletion on HIV-1 Env expression. However, since only Vpr null virus was used, it does not directly evaluate the contribution of Vpr-mediated PU.1 degradation on its capacity to enhance Env expression.

Reviewer Comment 5: SPI1 mRNA levels, encoding for PU.1 were also lower in the presence of Vpr (Figure 2B and 3B), suggesting that PU.1 downmodulation could occur at the transcriptional and post-translational level. It would be informative to determine the

contribution of the impact of Vpr on SPI1 expression on the reduced PU.1 protein levels. In that context, the impact of Vpr on endogenous PU.1 expression could be assessed +/- MG132.

Authors response:

We agree with the reviewer that Vpr likely acts at both the transcriptional and post-translational levels. To address the reviewer's concern, we performed a new set of experiments that clearly demonstrated degradation of endogenous PU.1 in primary macrophages following treatment of wild type HIV in a manner that was reversible by proteasome inhibitor treatment (MG132) as suggested (Figure 7 D, E). Thus, at least after short incubations (five hours), MG132 seemed to near-fully restore PU.1 levels. We do expect that after longer time points in primary cells that transcriptional mechanisms would lead to additional downmodulation of PU.1 in fully infected cells. Nevertheless, this experiment confirms the existing data in the manuscript performed in HEK 293T cells exogenously expressing both PU.1 and Vpr (Figure 7C).

Reviewer responses: These new data clearly demonstrate that Vpr promotes the degradation of PU.1 in MDMs.

Reviewer Comment 6: The authors claim that Vpr-mediated downmodulation of PU-1 regulated gene expression is mediated by direct Vpr-PU.1 interaction, resulting in accelerated proteasomal degradation of PU.1. They propose a model in which interactions amongst PU.1, Vpr and DCAF1 promote the ubiquitination of PU.1 via the associated CUL4A ubiquitin ligase complex with resultant proteasomal degradation. These conclusions are based on the data obtained with Vpr Q65R and DCAF1-depleted cells (Figure 7A). The Q65R substitution resulted in the loss of DCAF1 association and disrupted the formation of the Vpr- PU.1 complex (Figure 7A Lane 6). However, this mutation had no impact on Vpr-mediated PU.1 degradation (Lane 18 vs lane 14), how the authors explain this seemingly contradictory result? If Q65R is unable to recruit the E3 ligase and doesn't bind PU.1, then how can the authors explain the observed PU.1 degradation? (a quantification of PU.1 pulldown/ input, would be very helpful). Similarly, DCAF1 depletion reduced Vpr-PU.1 interaction (Lane 10), but had no impact on Vpr-mediated PU.1 degradation (Lane 22 vs lane 20). Again, how this result can be reconciled with the authors working model? Are these phenotypes linked to overexpression? Experiments aimed at measuring the impact of the Q65R mutation and DCAF1 depletion on PU.1 levels (in the context of endogenous PU.1 expression in MDMs) would strengthen the manuscript.

Authors response:

The reviewer's concern that overexpression of PU.1 might be limiting our ability to accurately detect degradation is very reasonable. In addition, the experimental protocol was optimized for immunoprecipitation rather than for assessing protein degradation. To assess degradation accurately, we found that it is important to utilize homogeneous cell cultures or single cell approaches (e.g. Figures 4 and 5) that can ensure expression of both Vpr and PU.1 in the sampled population. In addition to overexpressing PU.1, the cells used for immunoprecipitation contained heterogeneous mixtures of transfected and untransfected cells - including some transfected cells that contained PU.1 but not Vpr constructs. Thus, it is not surprising that western blot analysis of bulk cell lysates was not sufficiently sensitive to detect degradation in all experiments.

To address the reviewer's concern, we performed new experiments in primary macrophages as requested. The experiments were optimized to ensure that a majority of the cells in the culture were exposed to virion associated HIV Vpr. Under these conditions, we confirmed that Vpr promotes degradation of PU.1 through a mechanism that was inhibitable by the proteasome inhibitor MG132 (Figure 7 D, E). We also confirmed that degradation depended on an interaction between Vpr and DCAF1. This conclusion was based on the observation that the Vpr Q65R mutant that is defective at interacting with DCAF1 was also defective at promoting PU.1 degradation (Figure 7 A, B).

Reviewer response: I fully understand the author's explanation on the lack of sensitivity of PU.1 measurement in immunoprecipitation experiments. This could be discussed in the manuscript to explain this discrepancy. The new data showing the impact of Vpr on endogenous PU.1 levels in MDMs are more convincing and certainly strengthen the manuscript. They clearly demonstrate that Vpr promotes the degradation of PU.1 and that Vpr Q65R is defective. In view of these more convincing data with endogenous PU.1 in MDM, the authors should consider replacing the immunoprecipitation experiment with DCAF-1 depletion and Vpr Q65R in transfected cells (Figure 7F-G) by similar experiments in MDMs.

Reviewer #1 (Remarks to the Author):

Reviewer Summary: The current manuscript “HIV-1 Vpr combats the PU.1-driven antiviral response in primary human macrophages,” Virgilio and colleagues is well written, clear to follow and offers convincing evidence of the role of PU.1 in the activity of Vpr to down regulate innate defenses of macrophages infected with HIV-1.

This reviewer found the responses to the original critique to be adequate and recommend publication.

Author response: We appreciate the reviewer’s positive comments.

Comment #1.

One lingering issue that is discussed in the manuscript and to some length in the previous reviews is the effect of HIV-1 on bystander cells, and the suppression of innate defenses in bystander cells (defined as cells not making gag protein). Lim et al. (PMID: 35166645) published, using single cell analysis of HIV-1 infected THP-1 cells, that bystander cells respond to HIV by modifying their transcriptome. Further, they showed that macrophages containing pre-integration HIV-1 cDNA complexes exhibit different transcriptome profiles than control cells (not exposed to virus) or cells containing provirus. The question is: Would the possible presence of pre-integration HIV-1 cDNA complex containing cells, which would be included in the bystander group as defined by absence of gag, possibly complicate the interpretation of Vpr effects in bystander cells? I personally don’t think so, but I would appreciate the authors considering this possibility. In any case the Lim paper should be referenced since it first described HIV-1 effects on bystander cells using scRNA seq.

In the revised manuscript, we referenced the paper and added a discussion point in the text (Lines 450 – 459).

“A prior study reported scRNA-seq analysis of THP-1 cells transduced with a replication-defective HIV⁶⁸. The authors identified a population of cells with low HIV gene expression they felt was due to the presence of unintegrated pre-integration complexes (PIC) and noted transcriptomic changes within this population of cells compared to the fully infected population. Our study differed from this prior report in that we utilized wild type HIV infected primary macrophages and characterized Vpr-dependent transcriptomic changes comparing fully infected and uninfected (HIV-RNA-negative), bystander cells. We did not identify a similar population of cells that expressed low levels of HIV gene products. As their data was not made publicly available, we were unable to make a direct comparison between the data sets.”

Reviewer #2 (Remarks to the Author):

The revised manuscript by Virgilio is improved and provides additional evidence that PU.1 downregulation through a non-Vpr mediated method, can also enhance HIV Env expression. They also demonstrate that the Vpr in virus particles is sufficient to downregulate PU.1 in a proteasome-dependent manner. Both points strengthen an already compelling manuscript, shedding additional light on how Vpr enhances growth in monocyte macrophages.

Author response: We appreciate the reviewer’s positive comments.

Reviewer #3 (Remarks to the Author):

Author response: see below

Reviewer #4 (Remarks to the Author):

The authors have addressed the majority of the comments and provided detailed responses. Just a minor comment

- In Figure 1E, the authors should provide some marker genes/features for annotating the identified clusters of MDMs (Cluster 0: pro-inflammatory macrophages and Cluster 1: cycling cell).

Author response: We appreciate the reviewer's positive comments. We added the requested marker genes/features corresponding to cell-cycling for annotating the MDM clusters to the revised manuscript (Figure 1F).

Reviewer 3

Reviewer summary: In this study, Virgilio et al., used single cell RNAseq to capture the transcriptional changes mediated by Vpr during HIV-1-infection of macrophages. The authors previously demonstrated that Vpr counteracts accelerated degradation of the Env glycoprotein initiated by the mannose receptor (MR) by suppressing expression of the MR gene, MRC1. However, the mechanism by which Vpr suppresses MRC1 expression wasn't known. Here they show that Vpr reduces the levels of PU.1, a transcription factor regulating many essential macrophage genes, including MRC1. These timely results suggest a model in which Vpr could reprogram HIV-1-infected macrophages by targeting PU.1. Accordingly, Vpr was shown to disrupt the expression of several PU.1 associated antiviral factors, including MRC1, but also IFITM3, ISG15, IFI6 and LY96. Importantly, the capacity of Vpr to reduce PU.1 levels was conserved among HIV-1, HIV-2 and SIV.

Original Reviewer Comment 1: While data presented suggest that PU.1 and DCAF1 form a complex with Vpr, it remains unclear whether PU.1 degradation requires direct Vpr-PU.1 interaction and the recruitment of the DCAF1-Cul4 E3 ubiquitin ligase complex. Moreover, direct experiments connecting Vpr-mediated PU.1 degradation to its capacity to regulate gene expression and favor HIV replication in MDMs are needed.

Original Authors response:

To address the reviewer's concern, we performed additional experiments that directly connect Vpr-mediated PU.1 degradation with enhanced HIV replication in MDMs. This was accomplished through the design of an HIV vector that allowed silencing of PU.1 in Vpr-null HIV-infected macrophages. Using this vector, we found that silencing PU.1 led to increased HIV Env expression compared to macrophages infected with a Vpr-null HIV that contained a scrambled control shRNA (Figure 8). These data help support the model that Vpr acts through its effects on PU.1 to counteract factors that target HIV Env and thereby limit spread of HIV. The revised manuscript was modified to include a discussion of these new findings (Lines 532-534).

Other new experiments provided in the revised manuscript further strengthen these conclusions; for example, we extended results initially performed in cell line models to primary HIV infected macrophages. Specifically, we provide validation that Vpr forms a complex with PU.1 and promotes proteasomal degradation of PU.1 in primary HIV- infected macrophages (Figure 6 and 7). Additionally, see response to comment 2 below for other reviewer-requested studies validating our conclusions in primary macrophages.

Reviewer response to revision: The new data in MDMs confirming the capacity of Vpr to promote PU.1 degradation are convincing and certainly contribute to strengthen the current manuscript. However, direct experiments connecting Vpr-mediated PU.1 degradation to its

capacity to regulate gene expression and favor HIV replication in MDMs are still missing (see comment 2, 3 and 4 below).

Author response: We appreciate the reviewer's positive comments on our convincing new data that we agree strengthens the manuscript.

Original Reviewer Comment 2: In Figure 2E and 3C, the authors show that Vpr reduces the expression of several transcripts involved in TLR and IFN-I signaling, such as ISG15, IFITM3 and IFI6 that contribute to host innate immune response to HIV. How this impacts the level of the proteins encoded by these transcripts was not reported. Evaluation of the levels of the proteins encoded by these transcripts (in HIV-1 infected and uninfected bystander MDMs) would confirm the impact of Vpr on host innate immune response.

Original Authors response:

To address this concern, we performed quantitative confocal microscopy and determined that infected (multi-nucleated Gag-positive primary macrophages) had lower levels of PU.1-dependent innate immune response gene products than macrophages infected with a Vpr-null HIV (Figure 3D-I). Furthermore, we quantified the protein levels in uninfected bystander (Gag-negative) MDMs and found reduced IFITM3 and ISG15 associated with Vpr exposure. This result confirms our scRNA-seq findings suggesting a bystander effect of Vpr on uninfected, virally exposed cells (Figure 3F, I). New data showing that relatively short (five hour) exposure to virion-associated Vpr causes detectable levels of PU.1 degradation in macrophages (Figure 7A & B, D & E) helps explain this result.

Reviewer response to revision: Confocal microscopy does not represent a robust and sensitive technique to measure protein levels. Moreover, the quantification was done on only few cells (between 17 to 27 cells total). Flow cytometry and Western blotting represent more quantitative techniques to measure IFITM3, ISG15 and IFI6 protein levels and these techniques are already used by the authors in the current manuscript to measure PU.1 and MR protein levels in infected MDMs. The system used to measure the impact of Vpr on PU.1 protein levels in Figure 4 by flow cytometry, could therefore be also used to measure IFITM3, ISG15 and IFI6 protein levels. Alternatively, their protein levels could be assessed in infected MDM by Western blot using the condition used in Figure 7. Since Vpr-mediated PU.1 degradation is observed under these conditions, this should allow the authors to measure the impact of Vpr in IFITM3, ISG15 and IFI6 protein levels. Notably, the same group previously measured IFITM3 expression in infected MDMs by Western blot (Lubow *et al.*, 2020, eLife).

Author Response: In their original request, the reviewer asked that an "Evaluation of the levels of the proteins encoded by these transcripts (in HIV-1 infected and uninfected bystander MDMs) would confirm the impact of Vpr on host innate immune response". To address this, we analyzed protein levels via fluorescence confocal microscopy. This approach allowed us to respond to the reviewer's request that we examine both infected cells and bystander cells, see manuscript lines 198-209. To address the reviewer's new concern that too few cells were analyzed, we now provide additional data from a total of 79-100 cells for each infected condition. The results we obtained are very robust and successfully confirmed the single cell RNA data by assessing protein levels. We selected this approach because western blot analysis by its nature does not allow infected cells to be differentiated from the bystander cells as originally requested by the reviewer. The other reviewers were satisfied with this result, which fully confirmed the transcriptomic analysis.

The western blot approach shown in Figure 7 was useful for identifying direct targets of Vpr that are degraded shortly after exposure to viral particles. However, this method is not appropriate to measure the effect of Vpr on downstream targets that depend on PU.1 for their

transcription because these indirect effects require a longer time frame and Vpr from viral particles would not be expected to be stable for long enough for this assessment.

We attempted flow cytometric approaches, but antibody staining as assessed by flow for ISG15, IFITM3 was too dim and flow cytometry of HIV-infected multinucleated syncytia was problematic due to the large/heterogeneous cell size.

Original Reviewer Comment 3: The author's main conclusions are that Vpr reprograms HIV-1-infected MDMs gene expression by targeting PU.1 and that contributes to disrupt expression of several antiviral factors (MCR1, ISG15, FITM3, LY96 and IFI6) restricting HIV infection and spread. Evaluating the impact of PU.1 depletion on the capacity of Vpr to modulate antiviral gene expression in MDMs therefore favoring HIV replication would strengthen the conclusions of the manuscript.

Original Authors response:

As discussed above in our response to comment 1, prior studies have shown that Vpr is needed to counteract restriction of HIV Env expression and promote viral spread. In the revised manuscript we came full circle by showing that silencing PU.1 in HIV-Vpr-null- infected macrophages rescued HIV Env expression (Figure 8). This experiment was technically difficult because we and others have found that siRNAs and shRNAs can be sensed by macrophages leading to upregulation of antiviral factors that limit viral infection and spread. To avoid the si/shRNA response, which has limited our ability to infect siRNA- treated macrophages with HIV, we constructed an HIV that harbored shRNA cassettes (control scrambled shRNA and shRNAs directed against PU.1) in the Vpr open reading frame. This strategy generated sufficient infected cell numbers to conclude that silencing PU.1 rescued Env expression compared to control, which supports our model. However, evaluation of silenced primary macrophages to assess expression levels of antiviral factors in these cells was not feasible because all HIVs harboring shRNA cassettes, including the negative control, scrambled shRNA, did not spread well enough to acquire the cell numbers needed for this more extensive analysis.

Although we were unable to measure the impact of silencing PU.1 on interferon-induced genes (e.g. ISG15, IFITM3, IFI6) which are all upregulated in response to shRNAs, we instead examined whether silencing PU.1 alone could reduce mannose receptor, which is naturally highly expressed in macrophages and does not rely on innate immune sensing for upregulation. In this case, we were able to successfully demonstrate a reduction of mannose receptor with PU.1 silencing (Supplementary Figure 2B).

Reviewer response to revision: The data presented in Figure 8 are confirming the impact of PU.1 on HIV-1 Env and MR expression in MDMs. However, they do not directly address the impact of Vpr-mediated PU.1 degradation on its capacity to modulate antiviral gene expression and favor HIV replication, as these experiments were only done with a Vpr null virus. Previous studies by the same group (Mashiba *et al.*, 2014, Cell host Microbe; Collins *et al.*, 2015 Plos Pathog; Lubow *et al.*, 2020,eLife) and other groups (Klockow *et al.*, 2013, Virology; Zhao *et al.*, 2015 Nat Com; Kyei *et al.*, 2015, Cell Host Microbe; Krapp *et al.*, 2016, Cell Host Microbe) used si/shRNA to silence gene expression in MDMs prior HIV-1-infection. Notably, using shRNA-expressing lentiviral construct optimized to limit antiviral responses in MDMs, the same group previously silenced DCAF-1 and MR in MDMs before successfully infecting them with WT or Vpr- virus (Mashiba *et al.*, 2014, Cell host Microbe; Collins *et al.*, 2015 Plos Pathog; Lubow *et al.*, 2020,eLife) . Using this system, they determined the impact of DCAF-1 or MR silencing on the capacity of Vpr to enhance Env expression, virus production and replication. Similar experiments could be done here to directly evaluate the impact of PU.1 silencing on the capacity of Vpr to modulate antiviral gene expression and favor HIV-1 replication. To directly assess the impact of PU.1 degradation in the capacity of Vpr to favor HIV replication, the authors should measure viral spread (in MDMs or in coculture with T cells) or at least virion

production/infectivity as previously published (Mashiba *et al.*, 2014, Cell host Microbe; Collins *et al.*, 2015 Plos Pathog; Lubow *et al.*, 2020, eLife), not only Env expression.

Author Response: In their original request, the reviewer asked for “Evaluating the impact of PU.1 depletion on the capacity of Vpr to modulate antiviral gene expression in MDMs therefore favoring HIV replication”. This was achieved by silencing PU.1 in the infected cells using an HIV construct that also contained a cassette that silenced PU.1 and showing that this rescued Env expression, phenocopying Vpr. In addition, we showed that when we silenced PU.1 in macrophages (as shown in Supplemental Figure 2B) MRC1 was dramatically reduced.

As explained in the revised discussion (lines 490-496), this approach was used because initial strategies that targeted PU.1 using our previously published methods were tried and found to be not feasible. PU.1 is a master regulator necessary for macrophage differentiation. Thus, and as expected, silencing PU.1 prior to infection resulted in a change in the cellular phenotype to the point where the cells did not attach well to the plate and the cells were highly resistant to HIV infection, similar to what is observed with undifferentiated monocytes.

The strategy that we included in the paper, in which we constructed an HIV that contained a PU.1 knockdown cassette avoided this feasibility limitation and successfully confirmed that PU.1 knockdown phenocopied Vpr expression with respect to Env expression, which was the main point of the experiment.

Because we were unable to measure the impact of silencing PU.1 on interferon-induced genes (e.g. ISG15, IFITM3, IFI6), we tempered our conclusion that the effect of Vpr on these gene products was solely due PU.1 degradation as additional mechanisms are possible (see below and lines 496-500).

“While this strategy successfully confirmed that PU.1 knockdown increases Env expression, more research is needed to determine whether other Vpr-dependent transcriptional changes, such as those impacting the PU.1 and interferon-induced gene products ISG15, IFITM3 and IFI6, are mediated through Vpr-dependent degradation of PU.1 alone or whether additional Vpr-dependent pathways play a role.”

Original Reviewer Comment 4: The authors previously demonstrated that Vpr counteracts accelerated degradation of Env by reducing MCR1 expression. It is unclear why the authors didn't evaluate the impact of PU.1 depletion on Vpr's ability to reduce Env expression. Surprisingly, the only experiment evaluating the impact of Vpr on PU.1 levels in MDMs was performed using an Env-defective virus (Figure 4A,B,C). This looks like a straightforward experiment for this group and it would greatly benefit the manuscript.

Original Authors response:

As discussed above, this experiment was successfully performed in the revised manuscript (Figure 8).

Reviewer response to revision: This experiment indeed confirms the impact of PU.1 depletion on HIV-1 Env expression. However, since only Vpr null virus was used, it does not directly evaluate the contribution of Vpr-mediated PU.1 degradation on its capacity to enhance Env expression.

Author response: The reviewer's original request was to “evaluate the impact of PU.1 depletion on Vpr's ability to reduce Env expression.” This was a confusing request because Vpr promotes infection by increasing Env expression. Nevertheless, we interpreted this to mean that the reviewer wanted evidence that PU.1 depletion increased Env expression, phenocopying Vpr expression, which is what we successfully achieved as discussed in the prior comment.

Because both Vpr and PU.1 shRNA abolish PU.1 expression, the expected result of silencing PU.1 in a Vpr+ cells is no change - a negative result. Thus, the experiment that was done in which PU.1 silencing phenocopied Vpr and rescued Env expression was the more powerful approach and the one that succeeded and confirmed our single cell transcriptional, flow cytometric and confocal data.

Original Reviewer Comment 5: SPI1 mRNA levels, encoding for PU.1 were also lower in the presence of Vpr (Figure 2B and 3B), suggesting that PU.1 downmodulation could occur at the transcriptional and post-translational level. It would be informative to determine the contribution of the impact of Vpr on SPI1 expression on the reduced PU.1 protein levels. In that context, the impact of Vpr on endogenous PU.1 expression could be assessed +/- MG132.

Original Authors response:

We agree with the reviewer that Vpr likely acts at both the transcriptional and post- translational levels. To address the reviewer's concern, we performed a new set of experiments that clearly demonstrated degradation of endogenous PU.1 in primary macrophages following treatment of wild type HIV in a manner that was reversible by proteasome inhibitor treatment (MG132) as suggested (Figure 7 D, E). Thus, at least after short incubations (five hours), MG132 seemed to near-fully restore PU.1 levels. We do expect that after longer time points in primary cells that transcriptional mechanisms would lead to additional downmodulation of PU.1 in fully infected cells. Nevertheless, this experiment confirms the existing data in the manuscript performed in HEK 293T cells exogenously expressing both PU.1 and Vpr (Figure 7C).

Reviewer response to revision: These new data clearly demonstrate that Vpr promotes the degradation of PU.1 in MDMs.

Author response: We appreciate the reviewer acknowledging the strength of our new data.

Original Reviewer Comment 6: The authors claim that Vpr-mediated downmodulation of PU-1 regulated gene expression is mediated by direct Vpr-PU.1 interaction, resulting in accelerated proteasomal degradation of PU.1. They propose a model in which interactions amongst PU.1, Vpr and DCAF1 promote the ubiquitination of PU.1 via the associated CUL4A ubiquitin ligase complex with resultant proteasomal degradation. These conclusions are based on the data obtained with Vpr Q65R and DCAF1-depleted cells (Figure 7A). The Q65R substitution resulted in the loss of DCAF1 association and disrupted the formation of the Vpr- PU.1 complex (Figure 7A Lane 6). However, this mutation had no impact on Vpr-mediated PU.1 degradation (Lane 18 vs lane 14), how the authors explain this seemingly contradictory result? If Q65R is unable to recruit the E3 ligase and doesn't bind PU.1, then how can the authors explain the observed PU.1 degradation? (a quantification of PU.1 pulldown/ input, would be very helpful). Similarly, DCAF1 depletion reduced Vpr-PU.1 interaction (Lane 10), but had no impact on Vpr-mediated PU.1 degradation (Lane 22 vs lane 20). Again, how this result can be reconciled with the authors working model? Are these phenotypes linked to overexpression? Experiments aimed at measuring the impact of the Q65R mutation and DCAF1 depletion on PU.1 levels (in the context of endogenous PU.1 expression in MDMs) would strengthen the manuscript.

Original Authors response:

The reviewer's concern that overexpression of PU.1 might be limiting our ability to accurately detect degradation is very reasonable. In addition, the experimental protocol was optimized for immunoprecipitation rather than for assessing protein degradation. To assess degradation accurately, we found that it is important to utilize homogeneous cell cultures or single cell approaches (e.g. Figures

4 and 5) that can ensure expression of both Vpr and PU.1 in the sampled population. In addition to overexpressing PU.1, the cells used for immunoprecipitation contained heterogeneous mixtures of transfected and untransfected cells - including some transfected cells that contained PU.1 but not Vpr constructs. Thus, it is not surprising that western blot analysis of bulk cell lysates was not sufficiently sensitive to detect degradation in all experiments.

To address the reviewer's concern, we performed new experiments in primary macrophages as requested. The experiments were optimized to ensure that a majority of the cells in the culture were exposed to virion associated HIV Vpr. Under these conditions, we confirmed that Vpr promotes degradation of PU.1 through a mechanism that was inhibitable by the proteasome inhibitor MG132 (Figure 7 D, E). We also confirmed that degradation depended on an interaction between Vpr and DCAF1. This conclusion was based on the observation that the Vpr Q65R mutant that is defective at interacting with DCAF1 was also defective at promoting PU.1 degradation (Figure 7 A, B).

Reviewer response to revision: I fully understand the author's explanation on the lack of sensitivity of PU.1 measurement in immunoprecipitation experiments. This could be discussed in the manuscript to explain this discrepancy. The new data showing the impact of Vpr on endogenous PU.1 levels in MDMs are more convincing and certainly strengthen the manuscript. They clearly demonstrate that Vpr promotes the degradation of PU.1 and that Vpr Q65R is defective. In view of these more convicting data with endogenous PU.1 in MDM, the authors should consider replacing the immunoprecipitation experiment with DCAF-1 depletion and Vpr Q65R in transfected cells (Figure 7F-G) by similar experiments in MDMs.

Author response: In the original request, the reviewer stated that, "Experiments aimed at measuring the impact of the Q65R mutation and DCAF1 depletion on PU.1 levels (in the context of endogenous PU.1 expression in MDMs) would strengthen the manuscript".

We agree that where possible primary cell experiments are most physiologically relevant, and we have utilized these where it was feasible. However, working with HIV infected primary MDMs is also technically extremely challenging. For example, to perform the confirmatory IP showing Vpr and PU.1 interact in primary macrophages required 60 million primary macrophages per condition. Because the reviewers' concern related to overexpression of PU.1 in 293T cells, we opted to address the overexpression concern by performing experiments in K562 cells that express endogenous PU.1. Therefore, K562 cells allowed us to assess the contribution of both DCAF1 knockdown and Vpr-Q65R in cells endogenously expressing PU.1 as the reviewer originally requested in a more feasible system.

In addition, we added text to the manuscript addressing the apparent discrepancy in our ability to detect Vpr-mediated PU.1 degradation using western blot analysis of 293T cells overexpressing PU.1 (see below and lines 338-343 in manuscript).

"In comparison to single cell and flow cytometric approaches shown in Figures 4 and 5, which had the ability to differentiate infected from uninfected cells, western blot analysis of input lysates shown in Figure 6 had a low sensitivity to detect Vpr-mediated PU.1 degradation. This was likely due to overexpression of tagged exogenous PU.1, and heterogeneous mixtures of transfected cells that were not optimized to ensure that Vpr was expressed in all PU.1 positive cells."

REVIEWERS' COMMENTS

Reviewer #3 (Remarks to the Author):

I thank the authors for taking my comments seriously. As a result I believe that the manuscript is stronger and will provide an important contribution to the field.